# A $C^*$-algebraic view on the interaction of real- and reciprocal space topology in skyrmion crystals

Pascal Prass[1][*], Fabian R. Lux[2], Emil Prodan[2],
Duco van Straten[3] and Yuriy Mokrousov[1,4]

**1** Institute of Physics, Johannes Gutenberg University Mainz, 55099 Mainz, Germany
**2** Department of Physics, Yeshiva University, New York, NY 10016, USA
**3** Institute of Mathematics, Johannes Gutenberg University Mainz, 55099 Mainz, Germany
**4** Peter Grünberg Institut, Forschungszentrum Jülich, 52425 Jülich, Germany

[*] pprass@uni-mainz.de

## Abstract

Understanding the interaction of real- and reciprocal space topology in skyrmion crystals is an open problem. We approach it from the viewpoint of $C^*$-algebras and calculate all admissible Chern numbers of a strongly coupled tight-binding skyrmion system on a triangular lattice as a function of Fermi energy and texture parameters. Our analysis reveals the topological complexity of electronic states coupled to spin textures, and the failure of the adiabatic picture to account for it in terms of emergent electromagnetism. On the contrary, we explain the discontinuous jumps in the real-space winding number in terms of collective evolution in real-, reciprocal, and mixed space Chern numbers. Our work sets the stage for further research on topological dynamics in complex dynamic spin textures coupled to external fields.

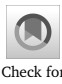

# 1   Introduction

Understanding the electronic properties of noncollinear magnetic systems is crucial for their implementation as novel, energy-efficient bits of information [1–3]. A rich class of noncollinear magnets is represented by the family of so-called magnetic multi-$q$ states: periodic arrangements of the local magnetic moments, dominated by their lowest order Fourier modes [4–6]. The homogeneity of these textures makes them ideal candidates to gain a deeper insight into the electronic structure of noncollinear magnetic materials and the interplay of topological features in real- and reciprocal space [7–15]. The multi-$q$ texture that holds the highest significance for the context of our work is the multi-$q$ skyrmion lattice. For example, skyrmionic states of this type appear in the family of chiral $B20$ materials such as MnSi [16, 17] and $Fe_{1-x}Co_xSi$ thin films [18, 19], chiral multiferroic insulators such as $Cu_2OSeO_3$ [20, 21], and have also been discovered in centrosymmetric materials such as $Gd_2PdSi_3$ [22] and $GdRu_2Si_2$ [23].

When the length scale of a skyrmionic multi-$q$ texture approaches the lattice constant of its host material, the resulting interference effects can lead to the formation of gapped topological states in the electronic system of the host material, manifesting for example, as a quantized topological Hall effect [7]. Traditionally, this is explained through the language of emergent magnetic fields [24–26]: using a unitary transformation, a noncollinear magnetic state can be rotated into a collinear co-moving frame of reference at the price of a modified kinetic energy which now resembles the presence of an electromagnetic vector potential. In the strong coupling limit of smooth textures, this analogy becomes exact with a corresponding "emergent" magnetic field, assuming the form

$$B_{\mathrm{em}} = \frac{\hbar}{2e}\hat{\mathbf{n}} \cdot (\partial_x \hat{\mathbf{n}} \times \partial_y \hat{\mathbf{n}})\,, \tag{1}$$

where $\hat{\mathbf{n}}: \mathbb{R}^2 \to \mathbb{R}^3$ with $\|\hat{\mathbf{n}}\| = 1$ describes a smooth skyrmion texture in two dimensions. This implies that a smooth multi-$q$ skyrmion system at strong coupling (i.e. in the adiabatic limit) can be mapped onto the Hofstadter model [27], which can explain the formation of a gap in the electronic spectrum and its topological nature [7, 28].

Finding a more general description of the effect is desirable. A suitable formalism would not rely on the adiabatic limit and, therefore, it would have a wider range of applicability [11]. In [10], we argue that all phenomena related to the presence of emergent fields are encoded in the observable algebra of the multi-$q$ system, irrespective of any assumption about adiabaticity.

An observable algebra is a so-called $C^*$-algebra [29] containing all translations of the Hamiltonian and other observables which can be represented as bounded, self-adjoint operators on a physical Hilbert space. For multi-$q$ systems, the observable algebra has a finite number of nontrivial equivalence classes of projections as described by operator $K$-theory [30]. Therefore, the gaps in the energy spectrum and their corresponding Fermi projections can be classified by a complete set of topological invariants including the ones responsible for quantized contributions to the Hall conductivity. The formalism in [10] can thus provide a topological classification of electronic states in multi-$q$ magnets without the need to refer to smooth magnetic textures or the adiabatic limit.

In this work, we demonstrate how to calculate all topological invariants of the observable algebra, which are real-, reciprocal, and mixed space Chern numbers, and determine them numerically for the triangular $3q$ skyrmion lattice over a range of texture parameters. In doing so, we obtain an exhaustive topological classification that uniquely identifies the $K$-theory classes of one of the standard models displaying a quantized topological Hall effect [7] [8, 9]. Our main findings are threefold: first, we find a single electronic band at the van-Hove singularity carrying the full contribution to the real-space Chern number. The second result concerns the relationship between the momentum space Chern number and the texture's real-space winding, which we find to be more intricate than traditionally assumed. Thirdly, we study the evolution of the electronic topology across various phase transitions of the magnetic texture in real-space and find that the observability of associated spectral flow is related to subtleties of the observable algebra. The existence of these gapless modes is related to the topology of the point defects which emerge at the transition parameter [31]. In summary, our work leverages the tools of noncommutative geometry to achieve a fine-grained understanding of the electronic states in multi-$q$ textures, which we believe to be key for the educated design of new topological materials based on magnetic structures.

The paper is organized into six sections. Section 2 formalizes the general mathematical properties of magnetic multi-$q$ textures, which form the basic ingredients for going into the formulation of the electronic observable algebra, presented in section 3. We focus on the real-space topological classification and demonstrate how parameter-dependent transitions between different topological phases occur. Section 3 identifies the observable algebra and its faithful representations, and we introduce a differential calculus, i.e., a trace and derivations, from which we construct the Chern characters. For numerical evaluation, the observable algebra and its differential calculus need to be approximated to the finite-volume case. This is done in section 4 together with a discussion of this approximation and its error bounds. With this computational method, the Chern numbers of gaps in the skyrmion crystal system are evaluated in section 5. We determine all Chern numbers for selected gaps and calculate the corresponding integer labels. This way, we verify their numerical precision and identify the Chern numbers of further interest. Next, we compute the main Chern numbers in dependence on the Fermi energy for selected texture parameters belonging to different topological phases of the texture. From this, we can identify which energy states contribute to which Chern number and how this depends on the texture's winding density. Finally, we compute the main Chern numbers for the Fermi energy in one fixed gap over a range of parameters, probing the transition between the texture's topological phases and compiling a phase diagram for the full range of texture parameters. From this, we identify the order parameters corresponding to the topological phase transition.

# 2 Properties of multi-$q$ magnetic systems

## 2.1 On the definition of multi-$q$ order

To begin with, we consider a general periodic texture $n: \mathbb{R}^d \to \mathbb{R}^3$ whose Bravais lattice can be constructed from the vectors $L_1, \dots, L_d$. In other words, these textures are invariant under the action of the discrete translation group $\{m_1 L_1 + \dots + m_d L_d | m \in \mathbb{Z}^d\} \subset \mathbb{R}^d$. Denote the basis for the corresponding reciprocal lattice by $Q_1, \dots, Q_d$, characterized by the property $L_i \cdot Q_j = 2\pi \delta_{ij}$. The resulting reciprocal lattice then comes in the form of

$$G_m = m_1 Q_1 + \dots + m_d Q_d\,, \tag{2}$$

with $m \in \mathbb{Z}^d$. As long as the texture $n$ is integrable within its unit cell, a lattice Fourier series can be given in the form of

$$n(x) = \sum_{m \in \mathbb{Z}^d} n_{G_m} e^{+iG_m \cdot x}\,, \tag{3}$$

$$n_{G_m} = \frac{1}{V_{uc}} \int_{V_{uc}} dx\, n(x) e^{-iG_m \cdot x}\,. \tag{4}$$

The decay rate of the Fourier coefficients $n_{G_m}$ as a function of increasing $\|m\|$ is controlled by the smoothness of the texture. A textbook example is the discontinuous square wave $n(x) = \mathrm{sgn}(\sin(x))$ in one dimension whose Fourier coefficients $n_m$ decay only as $1/m$. In contrast, for a sufficiently smooth texture, the Fourier decomposition will be dominated by its lowest order terms and one can take this as one hallmark of multi-$q$ magnetic order.

In this regime, a periodic magnetic texture can be thought of as a linear superposition of a few fundamental modes, fully characterized by a set of $r$ coexisting wave vectors $q_1, \dots q_r$ together with their initial phase conditions $\varphi_1, \dots, \varphi_r$ and amplitudes $n_{q_1}, \dots, n_{q_r}$ at the origin of some arbitrary, but fixed coordinate system:

$$n(x) \approx \sum_{i=-r}^{r} n_{q_i} e^{i(q_i \cdot x + \varphi_i)}\,, \tag{5}$$

with $q_{-i} := -q_i$, $n_{q_{-i}} := n_{q_i}^*$, and zero mode with $q_0 = 0$, $\varphi_0 = 0$, and $n_{q_0} \in \mathbb{R}^3$. The initial phase conditions are put in by design to clarify the intended phases at the origin and could be absorbed by the complex amplitudes. The $q$-vectors in the summation above can be any of the reciprocal lattice vectors $G_m$ with the coefficients $m_i \in \{-1, 0, 1\}$ (see Eq.(2)), i.e., the lowest order Fourier modes. Thus, $r$ may exceed the dimensionality $d$. In general, the vector field $n(x)$ has lost its normalization after the truncation of higher-order modes. A reconstruction of the normalization as $\hat{n}(x) = n(x)/\|n(x)\|$ still gives an excellent approximation to the original vector field albeit for the price of re-introducing higher order modes. Examples of multi-$q$ textures of this kind include the $A$-phase of MnSi which is characterized by three $q$-vectors [17], or the atomic scale skyrmion lattice in Fe/Ir(111) thin films which exhibits a superposition of two $q$-vectors [4]. The most fundamental example, however, is the stationary spin-wave: a single-$q$ state with only one mode, describing a plane wave of magnetic moments which manifests itself for example in the helimagnetic phases of MnSi [32–35] or FeGe [36,37].

One can now go as far as to take these stationary spin waves as the fundamental building blocks for more general magnetic textures by selecting arbitrary $q$-vectors from possibly different, and even outright incommensurate reciprocal lattices before assembling them in a linear superposition. This encompasses the Fourier truncation of lattice-periodic textures as

a special case but contains more general textures such as e.g. aperiodic Moiré patterns [38]. From now on, we parameterize these more general superpositions as

$$\boldsymbol{n}(\boldsymbol{x}) = \sum_{i=1}^{r} \left( \psi_i^c \hat{\mathbf{e}}_i^c \cos(\boldsymbol{q}_i \cdot \boldsymbol{x} + \varphi_i) + \psi_i^s \hat{\mathbf{e}}_i^s \sin(\boldsymbol{q}_i \cdot \boldsymbol{x} + \varphi_i) \right) + m\hat{\boldsymbol{e}}^0, \tag{6}$$

which can be seen to be the equivalent trigonometric formulation of Eq. (5) and which is more commonly used in the literature on multi-$q$ states, e.g., [6,38–40]. Here, $\psi_i^c$, $\psi_i^s$ are the amplitudes of the cosinusoidal and sinusoidal waves with the wave vectors $\boldsymbol{q}_i$, respectively, $\hat{\mathbf{e}}_i^c$ and $\hat{\mathbf{e}}_i^s$ are unit vectors, and $\hat{\boldsymbol{e}}^0$ is another independent unit vector for the direction of the constant net magnetization of magnitude $m$. In case of a normalized texture $\hat{\mathbf{n}}(\boldsymbol{x}) = \boldsymbol{n}(\boldsymbol{x})/\|\boldsymbol{n}(\boldsymbol{x})\|$, $m$ no longer determines the actual net magnetization but is still a related parameter. For example, one has $\lim_{m\to\infty} \hat{\mathbf{n}}(\boldsymbol{x}) = \hat{\boldsymbol{e}}^0$, the collinear ferromagnet.

A texture which is constructed this way from $r$ spin waves depends on the $r$ corresponding phases $\omega_i : \mathbb{R}^d \to S^1$, $\omega_i(\boldsymbol{x}) = (\boldsymbol{q}_i \cdot \boldsymbol{x} + \varphi_i) \mod 2\pi$. However, these phases are in general not independent of one another, as they are determined by the choice of the $q$-vectors. The phase values which can be reached at the same point in real-space lie in the image of the map $\boldsymbol{\omega} : \mathbb{R}^d \to T^r$. Only if one has $r$ $\mathbb{R}$-linear independent $q$-vectors every point in this phase space can be accessed, which is only possible for $r \leq d$. One can also consider the scenario where a selection of $q$-vectors is linearly dependent. The dimension of the image $\boldsymbol{\omega}(\mathbb{R}^d)$ is then not immediately apparent: let $\tilde{r}$ denote the dimension of the linear subspace of $\mathbb{R}^d$ spanned by the $r$ wave vectors. In case that the $\mathbb{R}$-linear dependent wave vectors are also linear dependent over $\mathbb{Q}$, there exists a rational relation between the phases and the texture is periodic. Then, the image $\boldsymbol{\omega}(\mathbb{R}^d) \subset T^r$ is of dimension $\tilde{r}$ and isomorphic to $T^{\tilde{r}}$. Otherwise, it may be dense in a submanifold of $T^r$ with dimension equivalent to the number of rationally independent wave vectors. This is the case for $q$-vectors from incommensurate lattices rendering the texture aperiodic. A simple example in $d = 1$ are two spin waves with phases of an irrational slope ratio, which wrap densely around $T^2$.

## 2.2 Real-space topological classification of multi-$q$ textures

Coming back to the case of periodic textures, there can be at most $d$ $\mathbb{Q}$-linear independent $q$-vectors and one has fixed rational relations between the initial phases. In the following the periodic case with $\tilde{r} = d$ is assumed. As the texture consists of a finite sum of smooth functions over $S^1$, it can be considered as an element of $C^\infty(T^d, \mathbb{R}^3)$, the space of smooth maps from the $d$-torus to $\mathbb{R}^3$. Whenever $\|\boldsymbol{n}(\boldsymbol{x})\| > 0$ for all $\boldsymbol{x} \in \mathbb{R}^d$, the normalisation procedure will lead to a smooth field of unit vectors $\hat{\mathbf{n}}$, and thus $C^\infty(T^d, S^2)$. By setting $\boldsymbol{x} = 0$, one can conveniently address $T^r$ with the corresponding $\varphi_i$ and restrict it to $T^d$ by using the phase relations.

The topological classification of these maps is based on the concept of homotopy. Two maps $f, g \in C(T^d, S^2)$ are called homotopic if there exists a continuous function $H : T^d \times [0,1] \to S^2$, such that $H(t,0) = f(t)$ and $H(t,1) = g(t)$ [41, Ch. 7]. Homotopy forms an equivalence relation on $C(T^d, S^2)$ with a corresponding quotient space denoted by $[T^d, S^2]$ - the space of homotopy classes. The most well-studied example of such quotients is the so-called homotopy groups of the sphere $\pi_n(S^k) = [S^n, S^k]$. In dimension two with two linear independent $q$-vectors, the elements of $[T^2, S^2]$ can be classified by their topological degree. Loosely speaking, the degree of a map $f$ counts the number of times the domain of $f$ is wrapped around the range of $f$. Since the texture $\hat{\mathbf{n}} \in C^\infty(T^2, S^2)$ is smooth in addition to being continuous (in fact, every element of $[T^2, S^2]$ is homotopic to a smooth map), the degree of $\hat{\mathbf{n}}$ can be

calculated as [42, Thm. 17.35]

$$\deg \hat{\mathbf{n}} = \frac{1}{4\pi} \int\limits_{T^2} \mathrm{d}\boldsymbol{\varphi} \ \hat{\mathbf{n}} \cdot (\partial_{\varphi_1} \hat{\mathbf{n}} \times \partial_{\varphi_2} \hat{\mathbf{n}}) \in \mathbb{Z}. \tag{7}$$

If one has two continuous maps $f, g \colon T^2 \to S^2$, then $\deg f = \det g$ if and only if $f$ and $g$ are homotopic – a result known as Hopf's theorem [43, p. 51]. Therefore, Hopf's theorem provides an isomorphism $[T^2, S^2] \cong \mathbb{Z}$ which provides a means of topological classification.

### 2.3 Topological phase transitions in real-space

Parameter manipulation, such as tuning the spin waves' phase relations $\varphi_i$ or varying the ferromagnetic degree of freedom $m$, changes the superposition in Eq. (6) smoothly. However, tuning the parameters over certain critical points can lead to destructive interference. In this case, $\|\boldsymbol{n}(\boldsymbol{x}_c)\| = 0$ will occur at critical positions $\boldsymbol{x}_c$ in real-space. The normalization procedure is thus undefined at $\boldsymbol{x}_c$ and no matter which choice will be made to redefine $\hat{\mathbf{n}}(\boldsymbol{x})$, the resulting function $\hat{\mathbf{n}}(\boldsymbol{x})$ will be discontinuous at the singularities $\boldsymbol{x} = \boldsymbol{x}_c$. This is of significant consequence, as the spectral properties of an electronic system coupled to this texture are sensitive to the topology of its singularities [31]. In practice, we can regularize the singularity by making the choice

$$\hat{\mathbf{n}} = \frac{\boldsymbol{n}}{\|\boldsymbol{n}\|} \longrightarrow \frac{\boldsymbol{n}}{\|\boldsymbol{n}\| + \eta} = \hat{\mathbf{n}}_\eta, \tag{8}$$

with $\eta > 0$. It is then guaranteed that $\lim_{\eta \to 0^+} \hat{\mathbf{n}}_\eta = \hat{\mathbf{n}}$ for all $\boldsymbol{x} \neq \boldsymbol{x}_c$.

We want to give a simple example that can be worked out exactly and that can serve as a toy model for more complex magnetic transitions. Consider the isolated skyrmion, parameterized in polar coordinates by

$$\boldsymbol{n}_{\mathrm{sk}}(\phi, \rho) = (\sqrt{1 - n_z(\rho)^2} \cos(\Phi(\phi)), \sqrt{1 - n_z(\rho)^2} \sin(\Phi(\phi)), n_z(\rho))^T, \tag{9}$$

where the polar angle of the vector direction $\Phi(\phi) = -\phi$ is opposite to the polar angle $\phi \in [0, 2\pi)$ of the coordinates, and the $z$-component $n_z(\rho) = (2\rho - 1)\Theta(1 - \rho) + \Theta(\rho - 1)$ with the Heaviside step function $\Theta$ and $\Theta(0) = \frac{1}{2}$ only depends on the radius $\rho \in [0, \infty)$, such that the texture is that of a ferromagnet for radius $\rho \geq 1$. To remain consistent with the framework of multi-$q$ textures, we can envision that this skyrmion occupies the cartesian unit cell $V_{uc} = [-1, 1] \times [-1, 1] \subset \mathbb{R}^2$. One has $\|\boldsymbol{n}_{\mathrm{sk}}\| = 1$ and a topological charge $\deg \boldsymbol{n}_{\mathrm{sk}} = 1$. We construct a linear interpolation to the ferromagnet by

$$\boldsymbol{n}(\phi, \rho, t) = (1 - t)\boldsymbol{n}_{\mathrm{sk}}(\phi, \rho) + t\boldsymbol{e}_z, \tag{10}$$

where $\boldsymbol{e}_z$ is the unit vector in $z$-direction. By design, a singularity occurs for $t = 1/2$ where we find $\boldsymbol{n} = 0$ at the origin. The resulting texture can now be regularized according to Eq. (8) before we calculate the winding number from formula Eq. (7). The integrals can be computed analytically. Let $t_c(\eta)$ represent the value of $t$ for which $\deg \hat{\mathbf{n}}_\eta$ has depleted to the value of $1/2$. As the notation indicates, the value of $t_c(\eta)$ will generally depend on $\eta$. However, one has $t_c(\eta) \to 1/2 \equiv t_c$ for $\eta \to 0$. Analytically, we then find the asymptotic scaling behavior (similar to [44, 45])

$$\deg \hat{\mathbf{n}}_\eta(t) \sim F\left((t - t_c)(\eta/\eta_0)^{-\kappa}\right), \quad \text{for } \eta \to 0^+, \tag{11}$$

where $\eta_0 = 1$, $\kappa = 1$, and where $F$ is the universal scaling function of the transition, explicitly given by

$$F(x) = \frac{1}{2}\left(1 - \frac{x}{1 + \frac{4x^2}{1 + 4|x|}}\right). \tag{12}$$

Moreover, one finds $\lim_{\eta \to 0^+} \deg \hat{\mathbf{n}}_\eta(t) = \Theta(t_c - t)$, for all $t \neq 1/2$. In summary, this example illustrates how topological invariants associated with a normalized multi-$q$ texture in real-space, such as its winding number, can change discontinuously as a function of the texture parameters [5, 6]. This is because, at the critical point, the texture ceases to be continuous, and its topological degree is not defined. The discontinuous transition can be turned into a continuous one at the price of momentarily leaving the function space $C(T^2, S^2)$ in favor of $C(T^2, \mathbb{R}^3)$, thus leaving the door open for a continuous change of the degree.

## 2.4  Multi-$q$ systems coupled to lattice electrons

Instead of the continuum $\mathbb{R}^d$, multi-$q$ textures are often studied from the perspective of an electronic tight-binding Hamiltonian which is hosted by its own respective Bravais lattice [7,8]. This Bravais lattice can generally be different from the Bravais lattice of the magnetic texture, and interesting band structure effects can result from the interference of these two patterns. Therefore, we choose a different notation for this case and define the $d$-dimensional electronic Bravais lattice by specifying its lattice vectors $\{\mathbf{a}_i\}$ and reciprocal lattice vectors $\{\mathbf{b}_j\}$, such that $\mathbf{a}_i \cdot \mathbf{b}_j = 2\pi \delta_{ij}$. The $r$ wave vectors $\{\mathbf{q}_i\}$ of the multi-$q$ texture can now be decomposed into $\mathbf{q}_i = \sum_j^d \theta_{ij} \mathbf{b}_j$. Accordingly, the real-space position of the lattice site with integer label $\mathbf{m} \in \mathbb{Z}^d$ is $\mathbf{x}_{\mathbf{m}} = \sum_j^d m_j \mathbf{a}_j$. This gives

$$\mathbf{x}_{\mathbf{m}} \cdot \mathbf{q}_i = \sum_{kl} m_k \theta_{il} \mathbf{a}_k \cdot \mathbf{b}_l = 2\pi \sum_k m_k \theta_{ik} \equiv 2\pi \, \mathbf{m} \cdot \boldsymbol{\theta}_i \,, \tag{13}$$

where $(\boldsymbol{\theta}_i)_k = \theta_{ik}$. This way, the coefficients $\theta_{ij}$ indicate by how much the phase $\omega_i(\mathbf{x}_{\mathbf{m}})$ of wave $i$ changes if one moves one site along $\mathbf{a}_j$. We denote the phase values modulo $2\pi$ by $[x] = x \mod 2\pi \in [0, 2\pi)$ for all $x \in \mathbb{R}$. Then, the phase factors can be written as

$$\omega_i(\mathbf{x}_{\mathbf{m}}) = [\mathbf{x}_{\mathbf{m}} \cdot \mathbf{q}_i + \varphi_i] = [2\pi \, \mathbf{m} \cdot \boldsymbol{\theta}_i + \varphi_i], \tag{14}$$

where $\varphi_i$ is the phase of wave $i$ at the origin $\mathbf{x}_0 = \mathbf{0}$ [10]. In matrix notation, one can give a more compact description in terms of

$$\boldsymbol{\omega}(\mathbf{x}_{\mathbf{m}}) = [2\pi \, \theta \mathbf{m} + \boldsymbol{\varphi}]. \tag{15}$$

The natural action $\tau$ of the lattice translation group $\mathbb{Z}^d$ on the waves phases is defined by

$$\boldsymbol{\omega}(\mathbf{x}_{\mathbf{m}+\mathbf{l}}) = [2\pi \theta(\mathbf{m} + \mathbf{l}) + \boldsymbol{\varphi}] = [\boldsymbol{\omega}(\mathbf{x}_{\mathbf{m}}) + 2\pi \theta \mathbf{l}] \equiv \tau_{\mathbf{l}} \boldsymbol{\omega}(\mathbf{x}_{\mathbf{m}}), \tag{16}$$

with $\mathbf{l} \in \mathbb{Z}^d$. If we take the right-hand side as the definition of $\tau$, we can relate all phase factors to the origin:

$$\boldsymbol{\omega}(\mathbf{x}_{\mathbf{m}}) = \tau_{\mathbf{m}} \boldsymbol{\omega}(\mathbf{x}_0) = \tau_{\mathbf{m}} \boldsymbol{\varphi} \,. \tag{17}$$

It is then possible to take a look at the entire orbit under lattice translations

$$\mathbb{Z}^d \boldsymbol{\varphi} = \{\tau_{\mathbf{m}} \boldsymbol{\varphi} \mid \mathbf{m} \in \mathbb{Z}^d\} \subset T^r \,, \tag{18}$$

and refer to the closure of $\mathbb{Z}^d \boldsymbol{\varphi}$ in $T^r$ as the hull $\Omega$ of the magnetic pattern. If the translation action is ergodic on $T^r$, e.g. for $r$ $\mathbb{R}$-linear independent lattice incommensurate wave vectors with $\theta_{ik} \in \mathbb{R} \setminus \mathbb{Q}$, then $\mathbb{Z}^d \boldsymbol{\varphi}$ will form a dense subset of $T^r$ whose closure is $\Omega = T^r$ for all $\boldsymbol{\varphi} \in T^r$. For the remainder of this work, we assume either immediately this case or we take on a periodic multi-$q$ texture with some rationally linear dependent lattice incommensurate wave vectors. Then, we can replace the dependent phases using the rational phase relations, thereby reducing $r$ to $\tilde{r}$, the number of $\mathbb{R}$-linear independent wave vectors, to return to the first case with ergodicity on $T^{\tilde{r}}$.

## 2.5 The hexagonal skyrmion crystal on the triangular lattice

The focus of this work is on the skyrmionic $3q$-state, supported by a triangular lattice. Its spin texture is given by the normalized superposition of $r = 3$ spin helices in two-dimensional real-space ($d = 2$) with wave vectors of equal length $q$ that sum up to zero (120° apart from each other) as illustrated in Fig. 1:

$$\boldsymbol{q}_1 = \left(\tfrac{\sqrt{3}q}{2}, \quad -\tfrac{q}{2}, \quad 0\right)^T, \qquad \boldsymbol{q}_2 = \left(0, \quad q, \quad 0\right)^T, \qquad \boldsymbol{q}_3 = \left(\tfrac{-\sqrt{3}q}{2}, \quad -\tfrac{q}{2}, \quad 0\right)^T. \tag{19}$$

The respective amplitudes are $1/\sqrt{3}$ and the unit vectors create a helical wave, i.e., $\hat{\mathbf{e}}_i^c$ directs out of the plane in the $z$-direction and $\hat{\mathbf{e}}_i^s$ is orthogonal to $\boldsymbol{q}_i$ and $\hat{\mathbf{e}}_i^c$, such that they form a right-handed basis of $\mathbb{R}^3$. The constant phase shifts in this scenario provide only one additional degree of freedom (up to real-space translation): their sum $\varphi_1 + \varphi_2 + \varphi_3 = \varphi$. The resulting vector field is given by the normalization $\hat{\mathbf{n}}$ of the superposition $\boldsymbol{n}$ [6]:

$$\boldsymbol{n}(\boldsymbol{x}) = \sum_{i=1}^{3} \left( \frac{1}{\sqrt{3}} \hat{\mathbf{e}}_z \cos(\boldsymbol{q}_i \cdot \boldsymbol{x} + \varphi_i) + \frac{1}{\sqrt{3}} (\hat{\boldsymbol{q}}_i \times \hat{\mathbf{e}}_z) \sin(\boldsymbol{q}_i \cdot \boldsymbol{x} + \varphi_i) \right) + m\hat{\mathbf{e}}_z \tag{20}$$

$$= \sum_{i=1}^{3} \left( \boldsymbol{n}_i e^{i(\boldsymbol{q}_i \cdot \boldsymbol{x} + \varphi_i)} + \boldsymbol{n}_i^* e^{-i(\boldsymbol{q}_i \cdot \boldsymbol{x} + \varphi_i)} \right) + m\hat{\mathbf{e}}_z, \tag{21}$$

with $\boldsymbol{n}_i = \frac{1}{2\sqrt{3}} (\hat{\mathbf{e}}_z - i\hat{\boldsymbol{q}}_i \times \hat{\mathbf{e}}_z)$. We consider this texture on a two-dimensional triangular lattice with real- and reciprocal space lattice vectors:

$$\boldsymbol{a}_1 = \left(1, \quad 0, \quad 0,\right)^T, \qquad\qquad \boldsymbol{a}_2 = \left(\tfrac{1}{2}, \quad \tfrac{\sqrt{3}}{2}, \quad 0\right)^T, \tag{22}$$

$$\boldsymbol{b}_1 = 2\pi\left(1, \quad -\tfrac{1}{\sqrt{3}}, \quad 0\right)^T, \qquad\qquad \boldsymbol{b}_2 = 2\pi\left(0, \quad \tfrac{2}{\sqrt{3}}, \quad 0\right)^T. \tag{23}$$

As required, it is $\boldsymbol{a}_i \cdot \boldsymbol{b}_j = 2\pi\delta_{ij}$. Expanding the $q$-vectors in the reciprocal lattice vector basis, we find

$$\boldsymbol{q}_1 = \frac{q\sqrt{3}}{4\pi} \boldsymbol{b}_1, \qquad \boldsymbol{q}_2 = \frac{q\sqrt{3}}{4\pi} \boldsymbol{b}_2, \qquad \boldsymbol{q}_3 = \frac{q\sqrt{3}}{4\pi}(-\boldsymbol{b}_1 - \boldsymbol{b}_1). \tag{24}$$

Let $\vartheta = \frac{q\sqrt{3}}{4\pi}$, then the decomposition coefficient matrix $\theta$ is given by:

$$\theta = \vartheta \begin{pmatrix} 1 & 0 \\ 0 & 1 \\ -1 & -1 \end{pmatrix}. \tag{25}$$

Of course, the third phase factor is artificial in the sense that three coplanar vectors cannot be linearly independent. As such, it can be eliminated from the whole description of the texture from the start with $r = 2$ and the minimal $\theta$-matrix given by $\theta = \vartheta\,\mathrm{id}_2$. However, the construction of the hexagonal skyrmion lattice is much simplified with the superfluous, yet helpful degree of freedom which can also assist in the interpretation of small angle neutron scattering data for example [16].

The degree of $\hat{\mathbf{n}} \in C^\infty(T^2, S^2)$, introduced in Eq. (7) is the first important order parameter characterizing the texture. In Fig. 2 (a), we show how $\deg\hat{\mathbf{n}}$ depends on $m$ and $\varphi$, thereby reproducing the result of [6, Fig. 6]. Implicit equations for the phase boundaries were already determined in [6, Eq. (29)]. Another relevant order parameter of $\hat{\mathbf{n}} \in C^\infty(T^2, S^2)$ for the interpretation of the electronic phase diagrams later on is the net magnetization

$$\langle \hat{\mathbf{n}} \rangle = \lim_{|V| \to \infty} \frac{1}{|V|} \int_V d^2\boldsymbol{x}\ \hat{\mathbf{n}}(\boldsymbol{\omega}(\boldsymbol{x})) = \frac{1}{|T^2|} \int_{T^2} d^2\boldsymbol{\varphi}\ \hat{\mathbf{n}}(\boldsymbol{\varphi}), \tag{26}$$

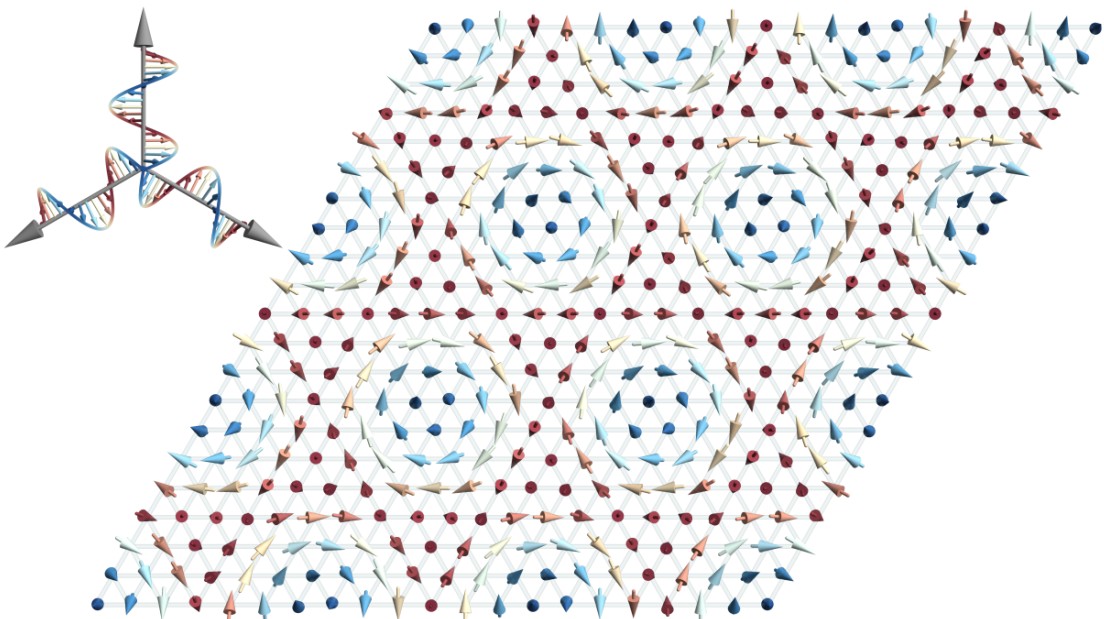

Figure 1: **Illustration of a multi-q state generating a skyrmion crystal.** Three helical spin waves with wave vectors summing up to zero in a two-dimensional plane, the initial phases at the origin are $\varphi_1 = \varphi_2 = \varphi_3 = 0$, generate a 3q-skyrmion texture with $\vartheta = 0.15$ on a triangular lattice.

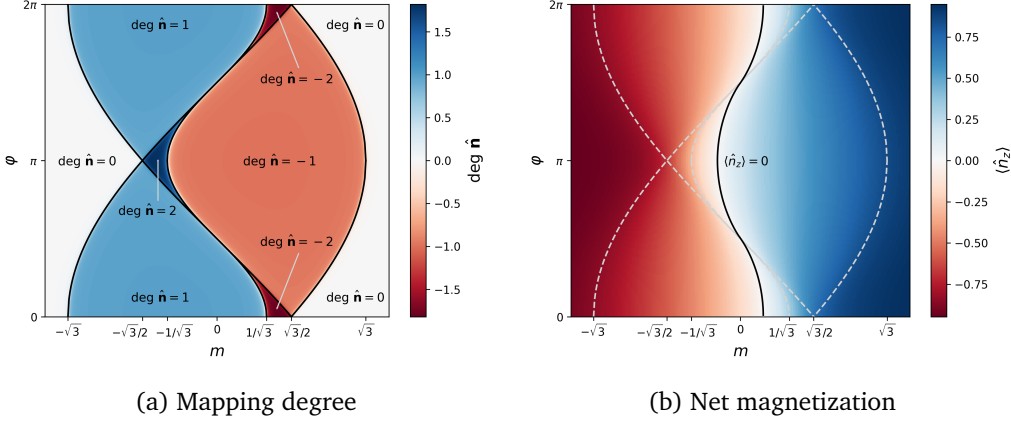

(a) Mapping degree

(b) Net magnetization

Figure 2: **Phase diagram of the 3q skyrmion lattice.** The subfigures show order parameters of the magnetic texture as a function of $m$ and $\varphi$ which are relevant for the interpretation of the electronic structure. (a) Gives $\deg \hat{\mathbf{n}}$, with phase boundaries (solid black lines) determined by the condition that $n(\varphi) = 0$ for some $\varphi \in T^2$ (cf. [6, Fig. 6]). (b) shows the net magnetization $\langle \hat{n}_z \rangle$ with phase boundary (solid black line) determined by the condition $\langle \hat{n}_z \rangle = 0$.

where the transition from real-space to the 2-torus of phases is understood in the sense of section 2.2. One finds $\langle \hat{n}_x \rangle = \langle \hat{n}_y \rangle = 0$, while $\langle \hat{n}_z \rangle$ shows a nontrivial dependence on $m$ and $\varphi$ as demonstrated by Fig. 2 (b). When $\langle \hat{\mathbf{n}} \rangle = 0$, the magnetic texture is time-reversal invariant modulo a proper Euclidean symmetry operation. Since this imposes additional constraints on the associated electronic system, we plot the resulting phase boundary as a solid black line in Fig. 2b).

Mathematically, important topological properties of the electronic system coupled to this magnetic texture can be discovered when $\vartheta$ takes irrational values. In this case, the map

$$\mathbb{Z}^2 \ni (m_1, m_2) \mapsto ([2\pi\vartheta m_1 + \varphi_1], [2\pi\vartheta m_2 + \varphi_2]) = (\omega_1, \omega_2) \in T^2, \tag{27}$$

samples the torus $T^2$ densely. To see that this is true, note the independence of $m_1$ and $m_2$ which are uniquely associated with either $\omega_1$ or $\omega_2$. We can therefore reduce the proof of the claim above to the following proposition: The set $\{[2\pi\vartheta m + \varphi] \mid m \in \mathbb{Z}\}$ is dense in $[0, 2\pi)$ for arbitrary $\varphi \in \mathbb{R}$ and $\vartheta \in \mathbb{R} \setminus \mathbb{Q}$. A proof can be found in [46] and [47, Prop. 1.3.3.]. For the two-dimensional skyrmion lattice just discussed, the hull of the magnetic pattern is thus given by

$$\overline{\{([2\pi\vartheta m_1 + \varphi_1, 2\pi\vartheta m_2 + \varphi_2]) \mid (m_1, m_2) \in \mathbb{Z}^2\}} \cong T^2. \tag{28}$$

As a consequence, the discrete — but infinite — electron system can "experience" all possible local realizations of the magnetic texture when $\vartheta$ is irrational. This will turn out to have very useful implications in the following.

## 3 The electronic observable algebra of multi-$q$ magnets

We claim in [10] that all possible tight-binding Hamiltonians incorporating multi-$q$ magnetic order are representations of a certain crossed product $C^*$-algebra $\mathcal{A}$ as bounded operators on the Hilbert space $\mathbb{C}^2 \otimes \ell^2(\mathbb{Z}^d)$. The algebraic structure of $\mathcal{A}$ encodes the essential physical properties of electrons in multi-$q$ magnets. In this section, we want to demonstrate why this is the case.

The emergence of magnetism in crystalline materials is a complex physical phenomenon, combining the special theory of relativity (in the form of the electron spin), quantum mechanics (due to the Pauli principle), and electromagnetism. On the level of Kohn-Sham density functional theory, a magnetic material comprising interacting electrons can be described by an effective non-interacting Hamiltonian [48, 49]

$$H_{KS} = -\frac{\hbar^2}{2m}\Delta + v_{\text{eff}}(\boldsymbol{r}) + \boldsymbol{B}_{\text{xc}}(\boldsymbol{r}) \cdot \boldsymbol{\sigma}, \tag{29}$$

where $v_{\text{eff}}(\boldsymbol{r})$ is the effective scalar potential, while $\boldsymbol{B}_{\text{xc}}(\boldsymbol{r})$ captures the exchange-correlation effects responsible for the emergence of magnetic order. We assume that $\boldsymbol{B}_{\text{xc}}(\boldsymbol{r})$ is a continuous vector field. This is the case, for example, in the local spin-density approximation to the exchange-correlation potential [49]. The basic toy model which describes some essential aspects of electrons in skyrmion materials can be either seen as a discretization of this Hamiltonian or as a projection onto local $s$-orbitals, given by the tight-binding Hamiltonian

$$H = -\lambda_{\text{hop}} \sum_{\langle \boldsymbol{i}, \boldsymbol{j} \rangle \in \mathbb{Z}^{2d}} |\boldsymbol{i}\rangle \langle \boldsymbol{j}| + \lambda_{\text{xc}} \sum_{\boldsymbol{i} \in \mathbb{Z}^d} \boldsymbol{\sigma} \cdot \boldsymbol{n}(\boldsymbol{x}_i) |\boldsymbol{i}\rangle \langle \boldsymbol{i}|, \tag{30}$$

where $\langle \boldsymbol{i}, \boldsymbol{j} \rangle$ indicates the restriction of the sum to nearest-neighbor pairs in the $d$-dimensional lattice. The parameter $\lambda_{\text{hop}}$ describes the hopping parameter, while $\lambda_{\text{xc}}$ describes the coupling strength to the magnetic texture $\boldsymbol{n}$ with $\boldsymbol{\sigma}$ representing the vector of Pauli matrices. The relevant Hilbert space is, therefore, $\mathcal{H} = \mathbb{C}^2 \otimes \ell^2(\mathbb{Z}^d)$, but we already want to stress that generalizations to further orbital and spin degrees of freedom, as well as next-nearest neighbor coupling and beyond, are possible for the formalism outlined below.

Here, we assume that the spectrum of the discrete tight-binding Hamiltonian exhibits the same topological features as the lowest part of the energy spectrum for its underlying continuum model. This has been confirmed in [50] for the case of a constant magnetic field perpendicular to a two-dimensional crystal in the strong binding regime, i.e., the integral quantum Hall effect.

For completely generic, continuous textures $\boldsymbol{n} \in C(\mathbb{R}^d, \mathbb{R}^3)$, the most general algebra which contains the Hamiltonian as one of its elements is a so-called von-Neumann algebra: $\mathcal{B}(\mathcal{H})$ – the space of bounded operators on $\mathcal{H}$. From the topological point of view, these algebras are too large to be useful: their $K$-theory is trivial, i.e., $K_0(\mathcal{B}(\mathcal{H})) = 0$ [30, p. 47]. For completely arbitrary textures, there is thus not much to be said. The case is different for multi-$q$ systems. This is because their associated textures can be thought of as a composition of maps [10]:

$$\mathbb{R}^d \xrightarrow{\;\boldsymbol{\omega}\;} T^r \xrightarrow{\;\boldsymbol{n}\;} \mathbb{R}^3 \,, \tag{31}$$

with $\boldsymbol{\omega}$ as in section 2.4 and where $\boldsymbol{n}$ is a continuous function. This structure can be leveraged to give a very compact description of an algebra of observables which is capable of describing every tight-binding electron Hamiltonian coupled to multi-$q$ magnetic order. Its definition and topological classification is the subject of this section: we rewrite the tight-binding Hamiltonian of Eq. (30) such that its affiliation to this algebra becomes apparent.

## 3.1 Crossed product algebra

The fundamental property that allows for a concise algebraic description is the fact that translations on the electronic lattice can be compensated by continuous shifts of the multi-$q$ texture — a property referred to as covariance in the mathematical literature. The resulting algebraic structures with built-in covariance are so-called crossed product $C^*$-algebras [51, Ch. 7], which is the content of this section.

As a first observation, the group of lattice translations $\mathbb{Z}^d$ has a corresponding continuous action on the algebra of continuous functions on the $r$-torus, $C(T^r)$, given via $C^*$-automorphisms:

$$\alpha \colon \mathbb{Z}^d \to \mathrm{Aut}(C(T^r)), \ \boldsymbol{m} \mapsto \alpha_{\boldsymbol{m}} \,, \tag{32}$$

$$(\alpha_{\boldsymbol{m}} f)(\varphi) \equiv f(\tau_{-\boldsymbol{m}} \varphi) = f(\varphi - 2\pi\theta \boldsymbol{m}) \,, \tag{33}$$

which we want to use to build the crossed product algebra. Recall that $\tau$ are the translations defined in Eq. (16). If one denotes the abstract generators of $\mathbb{Z}^d$ by $t_1, \ldots, t_d$, then one can think of $\mathbb{Z}^d$ as an abelian group with presentation

$$\mathbb{Z}^d = \langle t_1, \ldots, t_d \mid t_i t_j = t_j t_i \rangle \,. \tag{34}$$

We employ a multi-index notation and write $t^{\boldsymbol{m}} = t_1^{m_1} \ldots t_d^{m_d}$ for $\boldsymbol{m} \in \mathbb{Z}^d$. The group algebra $\mathbb{C}\mathbb{Z}^d$ then consists of formal summations $\sum_{\boldsymbol{m} \in \mathbb{Z}^d} c_{\boldsymbol{m}} t^{\boldsymbol{m}}$, where only finitely many coefficients $c_{\boldsymbol{m}} \in \mathbb{C}$ are different from zero and the multiplication law is inherited from the group structure by a linear extension.

As a vector space, the crossed product of the algebra $C(T^r)$ and the group $\mathbb{Z}^d$ is then defined as the $\mathbb{C}$-linear tensor product $C(T^r) \otimes \mathbb{C}\mathbb{Z}^d$ with a typical element in the form of

$$\sum_{\boldsymbol{m} \in \mathbb{Z}^d} f_{\boldsymbol{m}} \otimes t^{\boldsymbol{m}} \,, \tag{35}$$

and where the complex coefficients now have flourished into continuous functions on $T^r$, $f_{\boldsymbol{m}} \in C(T^r)$. Still, only finitely many $f_{\boldsymbol{m}}$ are different from zero. To obtain the algebraic structure on $C(T^r) \otimes \mathbb{C}\mathbb{Z}^d$ that is characteristic for the crossed product, one declares

$$(f_1 \otimes t^{\boldsymbol{m}_1})(f_2 \otimes t^{\boldsymbol{m}_2}) = f_1 \alpha_{\boldsymbol{m}_1}(f_2) \otimes t^{\boldsymbol{m}_1 + \boldsymbol{m}_2} \,. \tag{36}$$

We denote the resulting algebraic crossed product by $C_c(\mathbb{Z}^d, C(T^r), \alpha)$. While this description would already be sufficient to represent a wide class of model Hamiltonians $H$, the same cannot be said for their physical properties since these are encoded in the single-particle Green's

function, the resolvent $(z - H)^{-1}$ for $z \in \mathbb{C}$, which is not representable as an element of $C_c(\mathbb{Z}^d, C(T^r), \alpha)$. Therefore, we need a meaningful completion of the algebraic crossed product to a larger set of observables. This requires an additional structure that is introduced in the following section.

## 3.2 The noncommutative torus and its faithful representations

$C^*$-algebras have a long history in quantum mechanics [52] and provide a suitable framework to make sense of operators such as the resolvent and the Fermi projection. First, recall that a Banach algebra is an algebra $B$ over a field that is complete with respect to a norm $\|.\|$ fulfilling $\|ab\| \leq \|a\|\|b\|$ for all $a, b \in B$. A $C^*$-algebra $A$ is then a complex Banach algebra with an anti-multiplicative, anti-linear involution $*$, i.e., a Banach $*$-algebra, for which the so-called $C^*$-identity

$$\|a^*a\| = \|a\|^2,$$ (37)

is fulfilled for all $a \in A$. There are essentially two different ways to arrive at a $C^*$-completion of the algebraic crossed product $C_c(\mathbb{Z}^d, C(T^r), \alpha)$, which are referred to as the full crossed product $C^*$-algebra $C(T^r) \rtimes_\theta \mathbb{Z}^d$ and the reduced crossed product $C^*$-algebra $C(T^r) \rtimes_{\theta, r} \mathbb{Z}^d$. A detailed account can be found in [51, Ch. 7] while a brief, educational overview can be found in [53, Example 2.2.7].

Let $\mathcal{H}$ be a Hilbert space with its bounded operators denoted by $\mathcal{B}(\mathcal{H})$ and unitary operators denoted by $\mathcal{U}(\mathcal{H})$. A covariant representation $\pi \rtimes \rho$ of $C_c(\mathbb{Z}^d, C(T^r), \alpha)$ is a representation $\pi \colon C(T^r) \to \mathcal{B}(\mathcal{H})$ together with a unitary representation $\rho \colon \mathbb{Z}^d \to \mathcal{U}(\mathcal{H})$ such that

$$(\pi \rtimes \rho)(f \otimes t^m) = \pi(f)\rho(t^m),$$ (38)

$$\rho(t^m)\pi(f)\rho(t^{-m}) = \pi(\alpha_m(f)).$$ (39)

$C(T^r)$ itself is a $C^*$-algebra with involution given by the complex conjugation and with the uniform norm

$$\|f\|_{T^r} = \sup_{\varphi \in T^r} \|f(\varphi)\|.$$ (40)

We can use this norm to construct the $\ell^n$-norm of $a = \sum_{m \in \mathbb{Z}^d} f_m \otimes t^m$ as

$$\|a\|_{\ell^n} = \left( \sum_{m \in \mathbb{Z}^d} \|f_m\|_{T^r}^n \right)^{\frac{1}{n}}.$$ (41)

Denote by $\ell^1(\mathbb{Z}^d, C(T^r), \alpha)$ the Banach $*$-algebra which is obtained through the completion of $C_c(\mathbb{Z}^d, C(T^r), \alpha)$ with respect to the $\ell^1$-norm. The full crossed product $C^*$-algebra $C(T^r) \rtimes_\theta \mathbb{Z}^d$ is then obtained by completing $\ell^1(\mathbb{Z}^d, C(T^r), \alpha)$ with respect to

$$\|a\| = \sup_{\pi \rtimes \rho} \|(\pi \rtimes \rho)(a)\|_{\mathcal{B}(\mathcal{H})},$$ (42)

defined for all $a \in \ell^1(\mathbb{Z}^d, C(T^r), \alpha)$ and with the norm on the right being the operator norm, while the supremum is taken over all covariant representations $\pi \rtimes \rho$ of $C_c(\mathbb{Z}^d, C(T^r), \alpha)$.

Another $C^*$-algebra completion with closer ties to the physical description is the reduced crossed product $C^*$-algebra since it is built around the physical Hilbert space $\ell^2(\mathbb{Z}^d)$. For a generic element $a \in C_c(\mathbb{Z}^d, C(T^r), \alpha)$ given by $a = \sum_{m \in \mathbb{Z}^d} f_m \otimes t^m$, we define a family $\{\pi_\varphi^\rtimes\}_{\varphi \in T^r}$ of $*$-representations $\pi_\varphi^\rtimes = \pi_\varphi \otimes \rho$ by setting

$$\rho(t^m) = \hat{T}_m,$$ (43)

$$\pi_\varphi(f_m) = \sum_{i \in \mathbb{Z}^d} (\alpha_{-i} f_m)(\varphi) |i\rangle \langle i|,$$ (44)

where $\hat{T}_m$ is the unitary translation operator on $\ell^2(\mathbb{Z}^d)$ with $\hat{T}_m |i\rangle = |i + m\rangle$ for $i, m \in \mathbb{Z}^d$. The covariance is easily observed:

$$
\begin{aligned}
\rho(t^m)\pi_\varphi(f)\rho(t^{-m}) &= \sum_{i \in \mathbb{Z}^d}(\alpha_{-i}f)(\varphi)|i + m\rangle\langle i + m| \\
&= \sum_{i \in \mathbb{Z}^d}(\alpha_{-i}\alpha_m f)(\varphi)|i\rangle\langle i| \\
&= \pi_\varphi(\alpha_m f),
\end{aligned}
\tag{45}
$$

for all $f \in C(T^r)$ and $m \in \mathbb{Z}^d$. We can use $\pi_\varphi^\rtimes$ to formulate the norm

$$
\|a\| = \sup_{\varphi \in T^r}\|\pi_\varphi^\rtimes(a)\|_{\mathcal{B}(\mathcal{H})}.
\tag{46}
$$

The completion of $C_c(\mathbb{Z}^d, C(T^r), \alpha)$ with respect to this norm is the reduced crossed product $C^*$-algebra $C(T^r)\rtimes_{\theta,r}\mathbb{Z}^d$. For the noncommutative torus, it turns out that the two completions discussed in this section give identical $C^*$-algebras [54, Thm. 7.13], and we can drop the distinction in the notation from now on.

If the translation action on $T^r$ is ergodic, then $\pi_\varphi^\rtimes$ is faithful [55, Prop. 3.8]. This is because $\pi_\varphi^\rtimes(a) = 0$ if an only if $f_m(\tau_n\varphi) = 0$ for all coefficient functions $f_m$ and translates $n \in \mathbb{Z}^d$. Since the action is ergodic for all $\varphi \in T^r$ and the coefficient functions are continuous, one must have $f_m(\varphi) = 0$ everywhere. The faithfulness shows that our algebra does not contain any redundant or unnecessary information about the observables. It is the minimal algebraic structure that describes the essential physical properties of the electronic system: algebraically distinct observables in $C(T^r)\rtimes_\theta\mathbb{Z}^d$ will have physically distinct representations on $\mathcal{B}(\mathcal{H})$.

We can think of $C(T^r)$ in terms of the universal $C^*$-algebra

$$
C(T^r) = \langle u_1, \ldots, u_r \mid u_i u_j = u_j u_i \rangle,
\tag{47}
$$

where we represent the generators $u_i$ as the continuous phase factors $u_j(\varphi) = e^{i\varphi_j}$. The generators $u_j$ obey the following commutation relation in $C(T^r)\rtimes_\theta\mathbb{Z}^d$

$$
t_i u_j = \alpha_{e_i}(u_j)t_i = e^{-2\pi i\theta_{ji}}u_j t_i = e^{2\pi i(-\theta^T)_{ij}}u_j t_i,
\tag{48}
$$

which follows from Eq. (36). This means if we combine the generators of translations $t_i$ and the generators of continuous functions $u_j$ into a single notation:

$$
\gamma = (\gamma_1, \ldots, \gamma_{r+d}) = (t_1, \ldots, t_d, u_1, \ldots, u_r),
\tag{49}
$$

then we can give an equivalent description of $C(T^r)\rtimes_\theta\mathbb{Z}^d$ in terms of the universal $C^*$-algebra

$$
\mathcal{A}_\theta = C(T^r)\rtimes_\theta\mathbb{Z}^d = \langle\gamma_1, \ldots, \gamma_{r+d} \mid \gamma_i\gamma_j = e^{2\pi i\Theta_{ij}}\gamma_j\gamma_i\rangle,
\tag{50}
$$

where $\Theta$ is the $(r + d) \times (r + d)$ matrix

$$
\Theta = \begin{pmatrix} 0 & -\theta^T \\ \theta & 0 \end{pmatrix}.
\tag{51}
$$

In conclusion, the observable algebra of spin-1/2 electrons on the $\mathbb{Z}^d$ lattice in contact with an ergodic multi-$q$ texture is

$$
\mathcal{A} = M_2(\mathbb{C})\otimes(C(T^r)\rtimes_\theta\mathbb{Z}^d),
\tag{52}
$$

and we can extend $\pi_\varphi$ to $\mathcal{A}$ by setting

$$
\pi_\varphi(m \otimes a) = m \otimes \pi_\varphi(a),
\tag{53}
$$

for all $m \in M_2(\mathbb{C})$ and $a \in C(T^r)\rtimes_\theta\mathbb{Z}^d$.

### 3.3 The canonical form of the tight-binding Hamiltonian

To elucidate that the tight-binding Hamiltonian from Eq. (30) incorporating multi-$q$ magnetic order is indeed a representation of a noncommutative torus' algebra element, we rewrite the Hamiltonian into its canonical form. From a given site $\boldsymbol{i} \in \mathbb{Z}^d$, denote by $\boldsymbol{d}_1, \ldots, \boldsymbol{d}_c \in \mathbb{Z}^d$ the relative position of its nearest neighbors (which we assume to be independent of $\boldsymbol{i}$) with $c$ being the coordination number of the lattice. Further we define $\boldsymbol{\varphi} = \boldsymbol{\omega}(\boldsymbol{x_0})$. For the hopping term, one then finds

$$
H_{\text{hop}} = -\lambda_{\text{hop}} \sum_{\langle \boldsymbol{i}, \boldsymbol{j} \rangle \in \mathbb{Z}^{2d}} |\boldsymbol{i}\rangle \langle \boldsymbol{j}| = -\lambda_{\text{hop}} \sum_{\boldsymbol{i} \in \mathbb{Z}^d} \sum_{j=1}^c |\boldsymbol{i} + \boldsymbol{d}_j\rangle \langle \boldsymbol{i}| = -\lambda_{\text{hop}} \sum_{\boldsymbol{i} \in \mathbb{Z}^d} \sum_{j=1}^c \hat{T}_{\boldsymbol{d}_j} |\boldsymbol{i}\rangle \langle \boldsymbol{i}|
$$
$$
= -\lambda_{\text{hop}} \sum_{j=1}^c \pi_\varphi(t^{\boldsymbol{d}_j}). \tag{54}
$$

The algebra element

$$
\Delta = \sum_{j=1}^c t^{\boldsymbol{d}_j}, \tag{55}
$$

is also referred to as the adjacency operator of the lattice. The hopping term is now manifestly invariant under translations: $\hat{T}_{\boldsymbol{i}} H_{\text{hop}} \hat{T}_{\boldsymbol{i}}^\dagger = H_{\text{hop}}$. The behavior of the exchange term is slightly different. Since

$$
\boldsymbol{n}(\boldsymbol{\omega}(\boldsymbol{x}_{\boldsymbol{i}})) = \boldsymbol{n}(\tau_{\boldsymbol{i}}(\boldsymbol{\varphi})) = \alpha_{-\boldsymbol{i}}\boldsymbol{n}(\boldsymbol{\varphi}), \tag{56}
$$

it can be written as

$$
H_{\text{xc}} = \lambda_{\text{xc}} \sum_{\boldsymbol{i} \in \mathbb{Z}^d} \boldsymbol{\sigma} \cdot \boldsymbol{n}(\boldsymbol{\omega}(\boldsymbol{x}_{\boldsymbol{i}})) |\boldsymbol{i}\rangle \langle \boldsymbol{i}| = \lambda_{\text{xc}} \sum_{\boldsymbol{i} \in \mathbb{Z}^d} \alpha_{-\boldsymbol{i}}(\boldsymbol{\sigma} \cdot \boldsymbol{n})(\boldsymbol{\varphi}) |\boldsymbol{i}\rangle \langle \boldsymbol{i}|
$$
$$
= \lambda_{\text{xc}} \, \pi_\varphi(\boldsymbol{\sigma} \cdot \boldsymbol{n}). \tag{57}
$$

Combined, these results show that the Hamiltonian $H = H_{\text{hop}} + H_{\text{xc}}$ is given by the covariant representation $H = \pi_\varphi(h)$ with

$$
h = -\lambda_{\text{hop}} \, \Delta + \lambda_{\text{xc}} \, \boldsymbol{\sigma} \cdot \boldsymbol{n}, \tag{58}
$$

which is an element of $M_2(\mathbb{C}) \otimes (C(T^r) \rtimes_\theta \mathbb{Z}^d)$.

### 3.4 Traces on the noncommutative torus and Birkhoff's ergodic theorem

While it is intuitively clear what we mean by a trace on $\mathcal{B}(\mathcal{H})$, we have to do a little bit of work to unearth the same concept on the algebraic level of the noncommutative torus. As a starting point, there is a canonical trace on $C(T^r)$ given by

$$
\tau_{C(T^r)}(f) = \frac{1}{|T^r|} \int_{T^r} \mathrm{d}\boldsymbol{\varphi} \, f(\boldsymbol{\varphi}), \tag{59}
$$

with the useful property of being invariant under the $\mathbb{Z}^d$-automorphisms $\alpha$:

$$
\tau_{C(T^r)}(\alpha_{\boldsymbol{m}} f) = \tau_{C(T^r)}(f), \ \forall \boldsymbol{m} \in \mathbb{Z}^d. \tag{60}
$$

On the other hand, there is a unique trace on the group algebra $\mathbb{C}\mathbb{Z}^d$ given by

$$
\tau_{\mathbb{C}\mathbb{Z}^d}\left( \sum_{\boldsymbol{m} \in \mathbb{Z}^d} \gamma_{\boldsymbol{m}} t^{\boldsymbol{m}} \right) = \gamma_0. \tag{61}
$$

Combined, their composition can be used to define a trace $\mathcal{T}$ on the observable algebra $\mathcal{A}$ by declaring

$$\mathcal{T}\left(\sum_{\boldsymbol{m}\in\mathbb{Z}^d} f_{\boldsymbol{m}}\otimes t^{\boldsymbol{m}}\right) = \tau_{C(T^r)}(\text{tr } f_0), \tag{62}$$

where tr is the matrix trace of $M_2(\mathbb{C})$.

The physical relevance of the trace $\mathcal{T}$ is founded in the ergodic properties of the translation action on $T^r$ [55]. Let $\Pi_V$ be the projector onto a finite cube of volume $V$ centered around the origin. Using the covariance property of the $*$-representation $\pi_\varphi$, we then can write the trace per unit volume in terms of the summation

$$\mathcal{T}_V(a) = \frac{1}{|V|}\sum_{\boldsymbol{m}\in\mathbb{Z}^d} \text{tr } \langle\boldsymbol{m}|\,\Pi_V^\dagger \pi_\varphi(a)\Pi_V\,|\boldsymbol{m}\rangle = \frac{1}{|V|}\sum_{\boldsymbol{m}\in V} \text{tr } \langle 0|\,\hat{T}_{\boldsymbol{m}}^\dagger \pi_\varphi(a)\hat{T}_{\boldsymbol{m}}\,|0\rangle \tag{63}$$

$$= \frac{1}{|V|}\sum_{\boldsymbol{m}\in V} \text{tr } \langle 0|\,\pi_{\tau_{\boldsymbol{m}}(\varphi)}(a)\,|0\rangle . \tag{64}$$

By our assumption, $\tau\colon T^r \to T^r$ is ergodic. We then have a measure-preserving ergodic system $(T^r, \mathscr{B}, \mu, \tau)$, where $\mathscr{B}$ is the Borel $\sigma$-algebra of $T^r$, $\tau$ are the irrational rotations acting on $T^r$ and $\mu$ is the Lebesgue measure on $T^r$, invariant under $\tau$. In fact, $\tau$ is uniquely ergodic in the sense that the Lebesgue measure is the only $\tau$-invariant measure on $T^r$ [47, Def. 4.1.7.]. Uniquely ergodic systems fulfill a stronger version of Birkhoff's ergodic theorem [47, Prop. 4.1.13.] such that

$$\lim_{|V|\to\infty} \frac{1}{|V|}\sum_{\boldsymbol{m}\in V} \text{tr } \langle 0|\,\pi_{\tau_{\boldsymbol{m}}(\varphi)}(a)\,|0\rangle = \frac{1}{(2\pi)^r}\int_{T^r} \mathrm{d}\varphi\, \text{tr } \langle 0|\,\pi_\varphi(a)\,|0\rangle = \mathcal{T}(a), \tag{65}$$

converges uniformly for all $\varphi \in T^r$ and is independent of $\varphi$. In summary, we thus have

$$\lim_{|V|\to\infty} \mathcal{T}_V(a) = \mathcal{T}(a). \tag{66}$$

The trace per unit volume is perhaps the more natural trace to use and has a clear physical interpretation within the relevant physical Hilbert space. This result, therefore, shows that this trace is identical to its purely algebraic counterpart in the thermodynamic limit and is important to establish the physical significance of the algebraic formulation.

## 3.5 $K$-theory of the noncommutative torus

Operator $K$-theory is the noncommutative analog of topological $K$-theory classifying isomorphism classes of vector bundles. While the latter is only equipped to investigate projections of the $C^*$-algebra of matrix-valued continuous functions on locally compact Hausdorff spaces, operator $K$-theory can also classify projections in more difficult, noncommutative $C^*$-algebras, such as the noncommutative torus. The relevance to physical properties is directly inferred from this: the observable $C^*$-algebra describing a physical system contains all spectral (Fermi) projections, which can thus be labeled by topologically invariant numbers, the so-called Chern numbers. They are closely related to further physical quantities, e.g., transport behavior. The following overview about $K$-theory summarizes the relevant concepts for the context of this work and is based on the textbook account of [30].

Projections are one of the key subjects of $K$-theory. A projection in a $C^*$-algebra $A$ is a self-adjoint, idempotent element $p$, i.e. $p = p^* = p^2$. There are several ways to define equivalences of $C^*$-algebra elements. From topology we are familiar with *homotopy*, i.e., two elements are $a, b \in A$ are *homotopic* $a \sim_h b$, if there exists a continuous map $v : [0,1] \to A, t \mapsto v(t)$ with

$v(0) = a$ and $v(1) = b$. For the equivalence of projections, the image of this map needs to be a projection at every point. Two additional concepts for equivalence of projections are given by the *Murray-von Neumann equivalence* $p \sim q$ if there exists $v \in A$ with $vv^* = p$ and $v^*v = q$, and the *unitary equivalence* $p \sim_u q$, if there exists a unitary element $u$ in the unitarization $A^+$ with $q = upu^*$.

Passing to matrix algebras $M_n(\mathbb{C}) \otimes A$ with arbitrary but finite dimension $n \in \mathbb{N}$ one can coincide these notions of equivalence of projections from $A$. Let now $\mathcal{P}_\infty(A)$ be the limit of the union over all projections in all of these finite-dimensional matrix algebras. On this set, one can define the equivalence relation $p \sim_0 q$ for projections $p \in M_n(\mathbb{C}) \otimes A$ and $q \in M_m(\mathbb{C}) \otimes A$ if there exists $v \in M_{m,n}(\mathbb{C}) \otimes A$ such that $p = v^*v$ and $q = vv^*$. With the direct sum on $\mathcal{P}_\infty(A)$ given by

$$p \oplus q = \mathrm{diag}(p,q) = \begin{pmatrix} p & 0 \\ 0 & q \end{pmatrix}, \tag{67}$$

the set of equivalence classes $\mathcal{P}_\infty(A)/\sim_0$ is an abelian semi-group. The notion of an inverse element which is crucial in the definition of a group is missing as of now.

If the $C^*$-algebra $A$ is unital, we can straightforwardly generate the Grothendieck group from this semi-group to define $K_0(A)$. One can view this process as similar to the way the integers are constructed out of natural numbers by "inventing" negative numbers. $K_0(A)$ is the only group, up to a unique group homomorphism, with the property that any two homotopic projections in $M_n(\mathbb{C}) \otimes A$ correspond to the same well-defined group element in a way which is consistent with addition and the group's neutral element [30, Prop. 3.1.8]. If $A$ is not unital, one defines $K_0(A)$ as the kernel of the homomorphism $K_0(A^+) \to K_0(\mathbb{C})$, where $A^+ \cong A \oplus \mathbb{C}$ is the unitarization of $A$.

For the observable algebra of spin-1/2 electrons $\mathcal{A} = M_2(\mathbb{C}) \otimes \mathcal{A}_\theta$ with the noncommutative torus $\mathcal{A}_\theta = C(T^r) \rtimes_\theta \mathbb{Z}^d$ of dimension $r + d$ one finds [55, Prop. 4.2.4]

$$K_0(\mathcal{A}) = K_0(\mathcal{A}_\theta) = \mathbb{Z}^{2^{r+d-1}}, \tag{68}$$

independent on the matrix $\theta$. For more on the $K_0$-group for higher dimensional noncommutative tori, we refer to [56]. This means, that there are $2^{r+d-1}$ generators $[e_J]$, which can be uniquely labeled by the subsets of indices $J \subseteq \{t_1, \ldots, t_d, u_1, \ldots, u_r\}$ of even cardinality. The equivalence class of any projection from $P_\infty(\mathcal{A})$ can be uniquely decomposed into these generators with integer coefficients $n_J$:

$$[p]_0 = \sum_{\substack{J \subseteq \{t_1, \ldots, t_d, u_1, \ldots, u_r\} \\ |J| \text{ even}}} n_J [e_J]_0. \tag{69}$$

If $p$ is homotopically deformed, it stays inside its $K_0$ class, and the integer labels $n_J$ do not change. They are the complete set of topological invariants associated with the projection $p$. For a Fermi projection of a physical observable algebra, they label the energy gap of the spectrum in which the Fermi energy is located.

## 3.6 The noncommutative $n$-th Chern number

To access all the $K_0$ labels of a specific projection $p$ we can pair its corresponding $K_0$ equivalence class $[p]_0$ with a so-called *cyclic cocycle* [53, 57, 58]. They generalize the concept of a bounded trace by defining group morphisms from the $K_0$ group to $\mathbb{C}$. Cyclic $n$-cocycles $\eta$ on an algebra $A$ are multi-linear maps in $\mathrm{Hom}_\mathbb{C}(A^{\otimes(n+1)}, \mathbb{C})$ which fulfil the additional conditions of being

cyclic, $\lambda\eta = \eta$, and closed under the Hochschild boundary map, $b\eta = 0$, where

$$\lambda\eta(a_0,\ldots,a_n) = (-1)^n \eta(a_n, a_0, \ldots, a_{n-1}) \quad \text{and} \tag{70}$$

$$b\eta(a_0,\ldots,a_{n+1}) = \sum_{j=0}^{n}(-1)^j \eta(a_0,\ldots,a_j a_{j+1},\ldots,a_{n+1}) + (-1)^{n+1}\eta(a_{n+1}a_0,\ldots,a_n), \tag{71}$$

for $a_0,\ldots,a_{n+1} \in A$. They can also more conveniently be described as the characters of $n$-cycles $(\Omega, d, \int)$ over the algebra $A$, which consist of a graded, differential algebra $(\Omega, d)$, with $\Omega^0 \cong A$, and a closed, graded trace $\int$. This way, one obtains a cyclic $n$-cocycle of the form [57, III.1.$\alpha$. Prop. 4]:

$$\eta(a_0,\ldots,a_n) = \int a_0 da_1 \ldots da_n. \tag{72}$$

In particular, if we have a linear functional with the trace property, such as the trace of the noncommutative torus $\mathcal{T}$, as well as $n$ mutually commuting derivations $\partial_1,\ldots,\partial_n : A \to A$ such that $\mathcal{T}(\partial_j(a)) = 0$ for any $a \in A$ and $j \in \{1,\ldots,n\}$, then a cyclic $n$-cocycle is given by [53, Example 3.6.6]

$$\eta(a_0,\ldots,a_n) := \sum_{\sigma \in S_n} \text{sgn}(\sigma)\, \mathcal{T}(a_0 \partial_{\sigma(1)} a_1 \ldots \partial_{\sigma(n)} a_n), \tag{73}$$

where $S_n$ denotes the permutation group of $n$ elements.

In the case of the noncommutative torus, we can define derivations associated with the real-space and the phase space. For this, we consider an action of the Lie group $T^{r+d}$ on the $C^*$-algebra $\mathcal{A}_\theta = C(T^r) \rtimes_\theta \mathbb{Z}^d$. Let the action $\rho : T^{r+d} \to \text{Aut}(\mathcal{A}_\theta)$ act on the generators and their coefficients the following way

$$\rho_{\boldsymbol{\lambda}}(f)(\varphi) = f(\varphi + \xi), \tag{74}$$

$$\rho_{\boldsymbol{\lambda}}(t_i) = e^{-ik_i} t_i, \quad i = 1,\ldots,d, \tag{75}$$

with $\boldsymbol{\lambda} = (\boldsymbol{\xi}, \boldsymbol{k}) \in [0, 2\pi]^{r+d}$ and $f \in C(T^r)$. This can be extended to a point-wise norm continuous group of $*$-automorphisms on the noncommutative torus [55, Prop. 3.10.].

Elements of the algebra for which the automorphisms $\rho_{\boldsymbol{\lambda}}$ are $n$-times differentiable for $\lambda$ are called $n$-times differentiable as well. They form a linear subspace dense in $\mathcal{A}_\theta$. The dense subalgebra $\mathscr{A}_\theta$ of smooth, i.e., infinitely differentiable, elements is a Fréchet pre-$C^*$-algebra [58, Prop. 3.45]. Since its $C^*$-completion is the full noncommutative torus, the inclusion $i : \mathscr{A}_\theta \to \mathcal{A}_\theta$ induces an isomorphism of $K_0$-groups $K_0(i) : K_0(\mathscr{A}_\theta) \to K_0(\mathcal{A}_\theta)$ [58, Thm. 3.44]. This means the smooth noncommutative torus $\mathscr{A}_\theta$ has the same $K_0$ group as the full noncommutative torus $C^*$-algebra $\mathcal{A}_\theta$ itself.

Further, the pre-$C^*$-property implies that the algebra is stable under holomorphic functional calculus, which is crucial for our purpose. It guarantees to us that any Fermi projection $p_F$ of the Hamiltonian $h$ belongs to the smooth noncommutative torus, provided the Fermi energy $E_F$ lies within a gap of the Hamiltonians spectrum $\sigma(h)$. Let $E_F \notin \sigma(h)$ and $\Gamma_F$ be the contour in $\mathbb{C}$ which encircles $\sigma(h) \cap (-\infty, E_F)$ and does not intersect $\sigma(h)$. Then the Fermi projection $p_F$ may be defined by the Riesz projector

$$p_F = \frac{1}{2\pi i} \oint_{\Gamma_F} (z - h)^{-1} dz. \tag{76}$$

The derivations obtained from differentiating the torus action are unbounded, closed $*$-derivations of the form

$$\partial_j^u a = \partial_j^u \sum_{\boldsymbol{n} \in \mathbb{Z}^d} a_{\boldsymbol{n}} \otimes t_{\boldsymbol{n}} = \sum_{\boldsymbol{n} \in \mathbb{Z}^d} (\partial_{\varphi_j} a_{\boldsymbol{n}}) \otimes t_{\boldsymbol{n}}, \tag{77}$$

and

$$\partial_j^t a = \partial_j^t \sum_{\boldsymbol{n}\in\mathbb{Z}^d} a_{\boldsymbol{n}} \otimes t_{\boldsymbol{n}} = -i \sum_{\boldsymbol{n}\in\mathbb{Z}^d} n_j a_{\boldsymbol{n}} \otimes t_{\boldsymbol{n}}. \tag{78}$$

They mutually commute and the trace of the derivation of any algebra element is zero. Putting this all together with the proper prefactor and the internal spin degrees of freedom, we obtain the following Chern number from pairing the Chern character with a projection $p$ of the observable algebra $\mathcal{A} = M_2(\mathbb{C}) \otimes \mathcal{A}_\theta$ [59]:

$$Ch_J([p]_0) = \frac{(2\pi i)^{|J|/2}}{(|J|/2)!} \sum_{\sigma\in S_{|J|}} (-1)^\sigma \mathcal{T}\left( p \prod_{j\in J} \partial_{\sigma_j} p \right), \tag{79}$$

with $J \subseteq \{t_1,\dots,t_d,u_1,\dots,u_r\}$ and $|J|$ even. For the particular case of $J = \emptyset$, one has $Ch_\emptyset([p]_0) = \mathcal{T}(p)$. Since cyclic cocycles on an algebra induce group homomorphisms from its $K_0$-group's generators to $\mathbb{C}$, we can evaluate the Chern number of each projection's equivalence class by its linear decomposition into the $K_0$-groups generators and the evaluation of the Chern characters of the generators. For the projection class

$$[p]_0 = \sum_{\substack{I\subseteq\{t_1,\dots,t_d,u_1,\dots,u_r\} \\ |I|\ \text{even}}} n_I [e_I]_0, \tag{80}$$

we obtain the Chern number

$$Ch_J([p]_0) = \sum_{\substack{I\subseteq\{t_1,\dots,t_d,u_1,\dots,u_r\} \\ |I|\ \text{even}}} n_I Ch_J([e_I]_0). \tag{81}$$

The value of the Chern numbers on the generators of $K_0$ can be determined explicitly, and one finds [60, 61] $Ch_J([e_I]_0) = 1$ if $J = I$, $Ch_J([e_I]_0) = 0$ if $J \not\subseteq I$, and $Ch_J([e_I]_0) = \mathrm{Pf}(\Theta_{I\setminus J})$ otherwise. The operation Pf denotes the Pfaffian and $\Theta_{I\setminus J}$ is the representation of $\Theta$ in the reduced index set $I \setminus J$. These identities require compatibility of the order of indices established in Eq. (49) and the sign convention in Eq. (74) and (75). Evaluating the Chern numbers enables us to extract the corresponding $n_I$'s. For example, for the $3q$ skyrmion lattice with $\mathcal{A}_\theta = C(T^2) \rtimes_\theta \mathbb{Z}^2$ and $K_0(\mathcal{A}_\theta) = \mathbb{Z}^8$, we can consider

$$\theta = \vartheta \begin{pmatrix} 1 & 0 \\ 0 & 1 \end{pmatrix}, \tag{82}$$

and find

$$Ch_\emptyset = n_\emptyset - \vartheta n_{\{t_1,u_1\}} - \vartheta n_{\{t_2,u_2\}} - \vartheta^2 n_{\{t_1,t_2,u_1,u_2\}}, \tag{83}$$

$$Ch_{\{t_1,t_2\}} = n_{\{t_1,t_2\}}, \tag{84}$$

$$Ch_{\{t_1,u_1\}} = n_{\{t_1,u_1\}} - \vartheta n_{\{t_2,u_2\}} - \vartheta n_{\{t_1 t_2 u_1 u_2\}}, \tag{85}$$

$$Ch_{\{t_1,u_2\}} = n_{\{t_1,u_2\}}, \tag{86}$$

$$Ch_{\{t_2,u_1\}} = n_{\{t_2,u_1\}}, \tag{87}$$

$$Ch_{\{t_2,u_2\}} = n_{\{t_2,u_2\}} - \vartheta n_{\{t_1,u_1\}} - \vartheta n_{\{t_1 t_2 u_1 u_2\}}, \tag{88}$$

$$Ch_{\{u_1,u_2\}} = n_{\{u_1,u_2\}}, \tag{89}$$

$$Ch_{\{t_1,t_2,u_1,u_2\}} = n_{\{t_1,t_2,u_1,u_2\}}. \tag{90}$$

Moreover, the linear system of equations can be inverted to provide a mapping from the Chern numbers onto the integer invariants:

$$n_\emptyset = Ch_\emptyset + \frac{-\vartheta(Ch_{\{t_1,u_1\}} + Ch_{\{t_2,u_2\}}) + \vartheta^2(\vartheta - 3)Ch_{\{t_1,t_2,u_1,u_2\}}}{\vartheta - 1}, \tag{91}$$

$$n_{\{t_1,t_2\}} = Ch_{\{t_1,t_2\}}, \tag{92}$$

$$n_{\{t_1,u_1\}} = \frac{Ch_{\{t_1,u_1\}} + \vartheta Ch_{\{t_2,u_2\}} + \vartheta(\vartheta + 1)Ch_{\{t_1,t_2,u_1,u_2\}}}{1 - \vartheta^2}, \tag{93}$$

$$n_{\{t_1,u_2\}} = Ch_{\{t_1,u_2\}}, \tag{94}$$

$$n_{\{t_2,u_1\}} = Ch_{\{t_2,u_1\}}, \tag{95}$$

$$n_{\{t_2,u_2\}} = \frac{Ch_{\{t_2,u_2\}} + \vartheta Ch_{\{t_1,u_1\}} + \vartheta(\vartheta + 1)Ch_{\{t_1,t_2,u_1,u_2\}}}{1 - \vartheta^2}, \tag{96}$$

$$n_{\{u_1,u_2\}} = Ch_{\{u_1,u_2\}}, \tag{97}$$

$$n_{\{t_1,t_2,u_1,u_2\}} = Ch_{\{t_1,t_2,u_1,u_2\}}. \tag{98}$$

## 3.7 Physical interpretation of Chern numbers

The Chern number with the most well-known connection to physics is $Ch_{\{t_1 t_2\}}$, describing the intrinsic contributions to the anomalous Hall effect:

$$\sigma_{[xy]} = \frac{e}{h} Ch_{\{t_1 t_2\}}. \tag{99}$$

The $K$-theory analysis shows that for the skyrmion multi-$q$ texture considered in this work, $Ch_{\{t_1 t_2\}} = n_{\{t_1 t_2\}}$ is quantized to integer values as expected [7]. We argue, that $Ch_{\{u_1 u_2\}}$ describes the real-space winding number of the spin of the electronic states, and our numerical Chern number computation reveals, that a single band at the van-Hove singularity carries the whole contribution to it. Aside from $Ch_{\{t_1 t_2\}}$ and $Ch_{\{u_1 u_2\}}$, the remaining Chern numbers could be potentially relevant in a physics context.

For example, the integrated density of states represents the total number of electrons per unit volume (the charge carrier density), and according to the previous analysis, it obeys the polynomial equation

$$\text{IDS} = Ch_\emptyset = n_\emptyset - \vartheta n_{\{t_1,u_1\}} - \vartheta n_{\{t_2,u_2\}} - \vartheta^2 n_{\{t_1,t_2,u_1,u_2\}}, \tag{100}$$

where all coefficients $n_J$ are integers. The IDS is typically fixed by the chemistry of the respective material. Assume two insulators with different IDS harbor the same topological state in the sense that the Fermi projections belong to the same $K$-theory class. Then, due to the different chemistry of the materials, $\vartheta$ must change in order to fulfill the constraints imposed by the polynomial equation above. Imagine the IDS could be tuned by the application of a gate voltage $\phi$. If the system remains insulating, the only way to accommodate the change in IDS is a variation of $\vartheta$, i.e.,

$$\partial_\phi \vartheta = -\frac{\partial_\phi \text{IDS}}{n_{\{t_1,u_1\}} + n_{\{t_2,u_2\}} + 2\vartheta n_{\{t_1,t_2,u_1,u_2\}}}. \tag{101}$$

This means that the topological states enable a magnetoelectric behavior, controlled by the mixed space Chern numbers. The mixed space Chern numbers are further related to the existence of mixed space Berry curvature that also has transport manifestations such as the chiral Hall effect [62, 63] and orbital magnetization [64].

Lastly, we want to mention that the Chern numbers can play an important role in shaping the dynamical properties of topological magnetic materials, as shown in [65]. In essence, an adiabatic time-dependent modulation of the magnetic order can induce spin and charge currents of topological origin.

# 4 The approximating algebra for computation

The tight-binding Hamiltonian acts on the infinite-dimensional Hilbert space $\mathbb{C}^2 \otimes \ell^2(\mathbb{Z}^d)$ and is therefore impossible to implement explicitly for computation. The conventional solution to this problem is to impose periodic boundary conditions. Mathematically, this boils down to a description of the group $\mathbb{Z}$ as a direct limit of normal subgroups, for example,

$$\mathbb{Z} = p^0 \mathbb{Z} \triangleright p^1 \mathbb{Z} \triangleright p^2 \mathbb{Z} \triangleright \dots, \tag{102}$$

where $p$ is a natural number greater than 1. The respective quotients $\mathbb{Z}/p^k \mathbb{Z} = \mathbb{Z}_{p^k}$ are the integers modulo $p^k$ and are finite dimensional. Their direct limit is in general much larger than $\mathbb{Z}$, but the quotient morphisms $\mathbb{Z} \to \mathbb{Z}_{p^k}$ together with the universal property of the limit imply the existence of an injective group morphism onto a dense subspace of the direct limit [66, Chap. 4.1]. One can then be sure that model Hamiltonians taken from the group $C^*$-algebra of $\mathbb{Z}^d$ can be described in terms of a sequence of finite-dimensional, approximate Hamiltonians with periodic boundary conditions whose resolvent, density of states, etc., converge to the true bulk limit as $k \to \infty$ [66, Chap. 4.3].

If the $\theta$ matrix contains irrational values, it seems impossible at first to use the prescription of converging periodic boundary conditions. Still, we can take a sequence of rational approximations to $\theta$ such that the convergence can be equally ensured as $k \to \infty$ [55, Ch. 5]. This can be done, for example, by using an expansion of $\theta$ in terms of continued fractions [67]. In that case, the lattice group action does not act ergodically on the texture and the trace does not exactly correspond to the integral over the full phase space. Still, it gives a good numerical approximation for increasingly irrational approximations of $\theta$ where bigger and bigger periodic supercells sample the phase space sufficiently well. The algebra and differential calculus representation must be modified accordingly to accommodate these finite approximations. This will be discussed in the subsequent sections.

## 4.1 Finite-volume approximating algebra

We consider the periodic approximating algebras in the form of [55, Ch. 4.4]

$$\mathcal{A}_\theta^N = \langle \gamma_1, \dots, \gamma_{r+d} \mid \gamma_i \gamma_j = e^{2\pi i \Theta_{ij}} \gamma_j \gamma_i, \ \gamma_i^{N_i} = 1 \rangle, \tag{103}$$

where $N_i \in \mathbb{N}$. Consistency among the relations can only be achieved for certain quantized values of the flux matrix with respect to the system sizes $N_i$. This is because

$$\gamma_i = \gamma_j^{N_j} \gamma_i = e^{2\pi i \Theta_{ji} N_j} \gamma_i \gamma_j^{N_j} = e^{2\pi i \Theta_{ji} N_j} \gamma_i. \tag{104}$$

Taking the block structure of the flux matrix into account, the entries of the flux matrix need to fulfill the quantization condition

$$[\Theta_{ji} N_j] = 0 \Longleftrightarrow [\theta_{ij} N_j] = 0, \tag{105}$$

for $i = 1, \dots, r$ and $j = 1, \dots, d$. If the $\theta_{ij}$ coefficients are rational, they can respectively be expressed with co-prime integers $a_{ij}, b_{ij} \in \mathbb{N}$, such that $\theta_{ij} = \frac{a_{ij}}{b_{ij}}$. We can then always find a system size with $N_1, \dots, N_d$ to meet the above requirements.

For each real-space dimension, $j = 1, \dots, d$, there exists an $N_j$ such that for each $i = 1, \dots, r$ there is an $m_{ij} \in \mathbb{N}$ with $m_{ij} b_{ij} = N_j$. The smallest such $N_j$ is the least common multiple of the $b_{ij}$, $i = 1, \dots, r$. Then we have $N_j \theta_{ij} = m_{ij} a_{ij} \in \mathbb{N}$, in other words, $[\theta_{ij} N_j] = 0$.

For simplicity, we will only consider the case $N_j = N = 2L+1$ for all $j = 1, \ldots, d$ with $L \in \mathbb{N}$. Let $V_N$ denote the cube of length $N = 2L+1$ centered at the origin. The canonical Hilbert space is then $\ell^2(V_N)$ and we have a field $\{\hat{\pi}_\varphi^\rtimes\}_{\varphi \in T^r}$ of covariant $*$-representations $\hat{\pi}_\varphi^\rtimes : \mathcal{A}_\theta^N \to \mathcal{B}(\ell^2(V_N))$ with $\hat{\pi}_\varphi^\rtimes = \hat{\pi}_\varphi \rtimes \hat{\rho}$ given by

$$\hat{\rho}(t_i) = \hat{T}_i, \tag{106}$$

$$\hat{\pi}_\varphi(f) = \sum_{m \in V_N} (\alpha_{-m} f)(\varphi) |m\rangle \langle m|, \tag{107}$$

for all translation generators $t_i$ and all $f \in C(T^r)$. $\hat{T}_i$ is again the translation operator, but this time on the finite lattice with periodic boundary conditions. Having the super-cell balanced, i.e. $m \in V_n$ implying $-m \in V_N$, makes $\hat{\pi}^\rtimes$ a $*$-homomorphism. $\hat{\pi}_\varphi^\rtimes$ provides us with a finite-dimensional matrix, suitable for our numerical computations.

## 4.2 Approximate differential calculus

The $u$-derivations are straightforward to compute since they can be obtained directly from the partial derivatives as

$$\pi_\varphi^\rtimes(\partial_j^u a) = \sum_{n \in \mathbb{Z}^d} \sum_{m \in \mathbb{Z}^d} (\partial_{\varphi_j} a_n(\tau_{-m}\varphi)) |m\rangle \langle m| \hat{T}_n = \partial_{\varphi_j} \pi_\varphi^\rtimes(a), \tag{108}$$

for $a \in \mathscr{A}_\theta$, which we implement numerically using a finite-difference quotient. This recipe works in exactly the same way for the finite-volume approximating algebras $\mathcal{A}_\theta^N$ and their representations $\hat{\pi}_\varphi^\rtimes$.

The representations of the $t$-derivations for $\mathscr{A}_\theta$ are computed from the commutator with the position operator as

$$\pi_\varphi^\rtimes(\partial_j^t a) = i[\pi_\varphi^\rtimes(a), \hat{x}_j], \tag{109}$$

where $\hat{x}_j$ is the position operator acting on $\ell^2(\mathbb{Z}^d)$, i.e., $\hat{x}$ has the spectral decomposition

$$\hat{x} = \sum_{m \in \mathbb{Z}^d} m |m\rangle \langle m|. \tag{110}$$

This requires some care in carrying it over to the finite approximations. For $a \in \mathcal{A}_\theta^N$ one finds [55, Prop 4.26] that

$$\hat{\pi}_\varphi^\rtimes(\partial_j^t a) = -i \sum_{\lambda^{2L+1}=1} c_\lambda \lambda^{X_j} \hat{\pi}_\varphi^\rtimes(a) \lambda^{-X_j}, \tag{111}$$

where

$$c_\lambda = \begin{cases} \frac{\lambda^{L+1}}{1-\lambda}, & \text{if } \lambda \neq 1, \\ 0, & \text{if } \lambda = 1. \end{cases} \tag{112}$$

The result follows from the identity

$$n = \sum_{\lambda^{2L+1}=1} c_\lambda \lambda^n, \quad \text{for} \quad n \in \{-L, \ldots, L\}, \tag{113}$$

which is the discrete Fourier transformation (DFT) of the sequence of $2L+1$ complex numbers $c_{-L}, \ldots, c_L$ onto the sequence $-L, \ldots, L$.

The last missing component is now the appropriate trace for the finite-volume approximating observable algebra $\mathcal{A}^N = M_2(\mathbb{C}) \otimes \mathcal{A}_\theta^N$. It is given in analogy to the trace of the noncommutative torus by [55, Prop 4.26]

$$\hat{\mathcal{T}}(a) = \frac{1}{(2\pi)^r} \int_{T^r} \mathrm{d}\varphi \ \mathrm{tr} \ \langle 0 | \hat{\pi}_\varphi^\rtimes(a) | 0 \rangle \ . \tag{114}$$

Since the action on the finite-volume approximation algebra is not ergodic, we cannot use Birkhoff's ergodic theorem again to obtain the trace per unit volume. However, $T^r$ is still invariant under the action, which implies

$$\hat{\mathcal{T}}(a) = \frac{1}{(2\pi)^r} \int_{T^r} \mathrm{d}\varphi \ \frac{1}{|V_N|} \sum_{\boldsymbol{m} \in V_N} \mathrm{tr} \ \langle \boldsymbol{m} | \hat{\pi}_\varphi^\rtimes(a) | \boldsymbol{m} \rangle \ . \tag{115}$$

For our numerical evaluation, we omit the phase space average and merely compute the finite trace per unit volume

$$\hat{\mathcal{T}}_{V_N}^\varphi(a) = \frac{1}{|V_N|} \sum_{\boldsymbol{m} \in V_N} \mathrm{tr} \ \langle \boldsymbol{m} | \hat{\pi}_\varphi^\rtimes(a) | \boldsymbol{m} \rangle \ , \tag{116}$$

for one fixed phase space origin $\varphi$. In the limit of $N \to \infty$ ergodicity is restored and (116) converges to the trace in the noncommutative torus. The consequences of this approximation are discussed in section 5. We argue that the finite trace per unit volume is physically motivated, as long as there is no physical averaging mechanism.

### 4.3 The approximate $n$-th Chern number

In order to compute Chern numbers we first need to determine the representation of the Fermi projection $p_F \in \mathcal{A}^N$ as defined in Eq. (76). Since the representations $\hat{\pi}_\varphi^\rtimes$ are homomorphisms they commute with the functional calculus. Thus, we first represent our Hamiltonian $h \in \mathcal{A}^N$ from Eq. (58) as a finite $2N^d \times 2N^d$ matrix $\hat{\pi}_\varphi^\rtimes(h)$, diagonalize it, and use the spectral theorem to construct the finite Fermi projection

$$P_F^\varphi = \hat{\pi}_\varphi^\rtimes(p_F) = \sum_{i=1}^{2N^d} \chi_{(-\infty, E_F)}(\epsilon_i) | \epsilon_i \rangle \langle \epsilon_i | \ , \tag{117}$$

where $\epsilon_i$ and $| \epsilon_i \rangle$, $i = 1, \dots, 2N^d$, are the energy eigenvalues and eigenvectors of $\hat{\pi}_\varphi^\rtimes(h)$ in $\mathbb{C}^2 \otimes \ell^2(V_N)$. The approximate derivations of the Fermi projection can be computed from this representation as established in section 4.2 and the approximate Chern number we obtain numerically is then computed as

$$\hat{Ch}_J^\varphi(p_F) = \frac{(2\pi i)^{|J|/2}}{(|J|/2)!} \sum_{\sigma \in S_{|J|}} (-1)^\sigma \frac{1}{|V_N|} \sum_{\boldsymbol{m} \in V_N} \mathrm{tr} \ \langle \boldsymbol{m} | \hat{\pi}_\varphi^\rtimes(p_F) \prod_{j \in J} \hat{\pi}_\varphi^\rtimes(\partial_{\sigma_j} p_F) | \boldsymbol{m} \rangle \ . \tag{118}$$

For readability, we will refer to the numerically evaluated Chern numbers of a specified gap in section 5 by $Ch_J$.

### 4.4 Error bounds

In [55, Chap. 6, 8] error bounds are established for the smooth and non-smooth correlation functions evaluated with the finite-volume approximation. These are valid for the finite-volume approximate algebra with its proper trace. Thus they are only applicable to the phase space average of our results. Nevertheless, we observe a matching convergence.

Let $h$ be the Hamiltonian from Eq. (58), $E_F$ the Fermi energy located in the spectral gap of $h$, and $p_F$ the corresponding Fermi projection as given in Eq. (76). Then for any $K \in \mathbb{N}$, there exists a finite constant $C_K$ such that

$$|Ch_J(p_F) - \frac{1}{(2\pi)^r} \int_{T^r} d\boldsymbol{\varphi} \; \hat{C}h_J^\varphi(p_F)| \leq \frac{C_K}{(1 + |V_N|)^K} \,, \tag{119}$$

where $J$ is any subset of indices $J \subseteq \{t_1, \ldots, t_d, u_1, \ldots, u_r\}$ of even cardinality. Thus, the convergence to the thermodynamic limit of the finite-volume approximation with phase space average is faster than any inverse power of the finite supercell's size.

## 5 Results

### 5.1 Full topological classification of electronic states in the hexagonal SkX phase

Topological features in the electronic structure are known to appear in the strong coupling limit $\|\lambda_{\text{xc}}/\lambda_{\text{hop}}\| \gg 1$ of two-dimensional skyrmionic crystals in the tight-binding approximation [7]. This has been the motivation for our previous work [10]. In this section, we want to complete our program for the hexagonal skyrmion crystal hosted by a triangular lattice as initiated in [10] and give a full list of all topological invariants that can be attributed to the observable gaps in the spectrum.

We first want to comment on which topological invariants can be expected to arise from very general arguments. Based on the discussions of the previous sections, we can view $h$ in this case as an element of the noncommutative torus $C(T^2) \rtimes_\theta \mathbb{Z}^2$ tensored with the operators $M_2(\mathbb{C})$ that act on the internal spin degree of freedom and write

$$h/\lambda_{\text{xc}} = (1 - 2p) + \epsilon \Delta \,, \tag{120}$$

where we assume $\lambda_{\text{xc}} > 0$, $\epsilon = \lambda_{\text{hop}}/\lambda_{\text{xc}}$, and where $\Delta$ is the adjacency operator of the lattice (the operator which couples nearest neighbors with unit strength). The operator $p$ depends on the magnetic texture $\hat{\mathbf{n}} \in C^\infty(T^2, S^2)$ and is given by the projection

$$p = \frac{1}{2}\left(1 \otimes t^0 - \boldsymbol{\sigma} \cdot \hat{\mathbf{n}}(\boldsymbol{\varphi}) \otimes t^0\right) \,. \tag{121}$$

When $\epsilon = 0$, the Hamiltonian $h$ has a gap at zero energy, referred to as the zero energy gap in the following. For $0 < \|\epsilon\| \ll 1$, this gap stays open, and the gap projection operator changes continuously as a function of $\epsilon$. For a topological classification of the zero energy gap, we can thus focus on the limit of $\epsilon \to 0$. The occupied subspace below zero energy belongs to the eigenspace of $p$ with an eigenvalue equal to one, characterized by a perfect anti-alignment of the electron spin with the vector field specified by $\hat{\mathbf{n}}$. We compute the Chern numbers

$$n_\emptyset = C_\emptyset([p]_0) = 1 \,, \tag{122}$$

and

$$n_{\{u_1, u_2\}} = C_{\{u_1, u_2\}}([p]_0) = -2\pi i \int_{T^2} \frac{d\boldsymbol{\varphi}}{(2\pi)^2} \; \text{tr} \, (\boldsymbol{\sigma} \cdot \hat{\mathbf{n}})[(\boldsymbol{\sigma} \cdot \partial_{\varphi_1} \hat{\mathbf{n}}), (\boldsymbol{\sigma} \cdot \partial_{\varphi_2} \hat{\mathbf{n}})]/8 \tag{123}$$

$$= \frac{1}{4\pi} \int_{T^2} d\boldsymbol{\varphi} \; \hat{\mathbf{n}} \cdot (\partial_{\varphi_1} \hat{\mathbf{n}} \times \partial_{\varphi_2} \hat{\mathbf{n}}) \tag{124}$$

$$= \deg \hat{\mathbf{n}} \,. \tag{125}$$

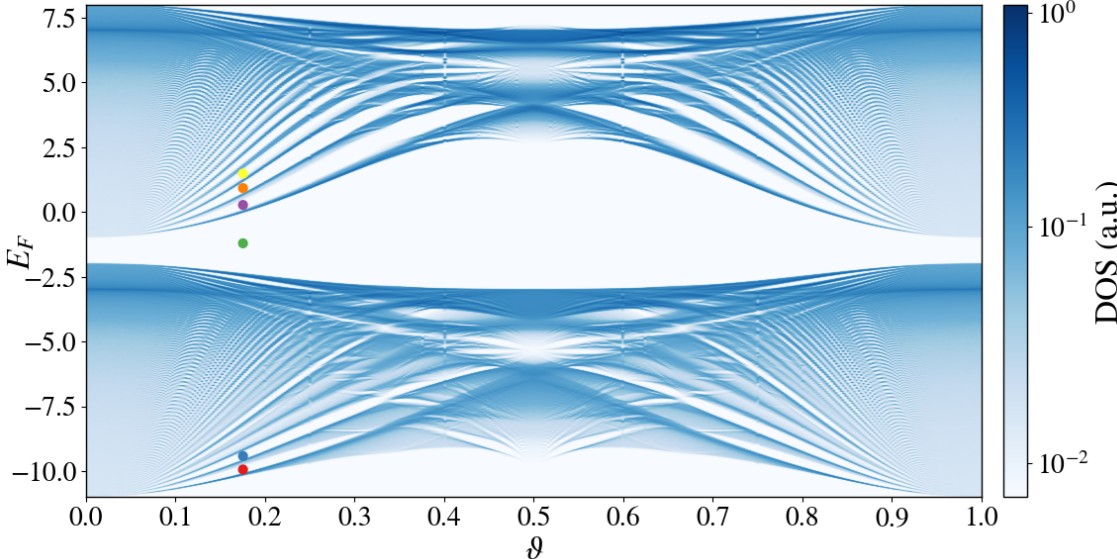

Figure 3: **Electronic spectrum for the 3$q$ skyrmion crystal.** The density of states of the tight-binding Hamiltonian coupled to the 3$q$ skyrmion crystal is computed over the entire range of length scales $\vartheta$.

The remaining Chern numbers for this gap are zero since $p$ commutes with the position operator. As the trivial projection $\mathbf{1}$ has the trace 2 and exhibits 0 for all other Chern numbers, we can also infer that the unoccupied subspace above zero energy, which is the eigenspace of

$$p^{\perp} = \mathbf{1} - p = \frac{1}{2}\left(\mathbf{1} \otimes t^0 + \boldsymbol{\sigma} \cdot \hat{\mathbf{n}}(\boldsymbol{\varphi}) \otimes t^0\right), \tag{126}$$

carries $n_{\emptyset} = 1$, $n_{\{u_1, u_2\}} = -\deg \hat{\mathbf{n}}$, and 0 for the remaining Chern numbers.

Another interesting quantity to look at is the spin magnetization. It has the expectation value

$$\boldsymbol{S} = \mathcal{T}(\boldsymbol{\sigma}p) = -\frac{1}{(2\pi)^2}\int\limits_{T^2} \mathrm{d}\boldsymbol{\varphi}\ \hat{\mathbf{n}}(\boldsymbol{\varphi}), \tag{127}$$

in units of $\hbar/2$. This reproduces Eq. (26) from section 2.5 and the spin magnetization will follow the phase diagram as shown in Fig. 2.

When the length scale of the skyrmion crystal now approaches the lattice constant, further energy gaps open inside the two blocks of aligned and anti-aligned states over and under zero energy. This can be observed by computing the spectrum of the tight-binding Hamiltonian for a range of $\vartheta$ values via exact diagonalization or more sophisticated algorithms such as the *Kernel Polynomial Method* (KPM) [68] which makes it computationally viable to evaluate much larger supercells. In Fig. 3, we used the KPM to evaluate the spectrum of the tight-binding Hamiltonian from Eq. (30) with parameter $\lambda_{\mathrm{hop}} = 1$ and $\lambda_{\mathrm{xc}} = 5$ on a supercell of size $N_x = N_y = 400$. The magnetic texture $\hat{\mathbf{n}} \in C^{\infty}(T^2, S^2)$ is implemented as in Eq. (20) with phase shift and magnetization parameters $\varphi_1 = \varphi_2 = \varphi_3 = 0$ and $m = 0$. For the KPM algorithm, we choose 2048 energy values, 2048 Chebyshev moments, and 10 random states. The length scale parameter $\vartheta$ takes values in a system of size $N = 400$, i.e., $\vartheta = \frac{i}{400}$ with $i = 1, \ldots, 199, 201, \ldots, 399$.

Table 1: **Full topological classification of the 3$q$ skyrmion crystal.** For the gaps labelled by the colored bullet points in the density of states in Fig. 3, we compute all possible Chern numbers at $\vartheta = \frac{10}{57}$ in Table (a) and extract the corresponding $K$-theory class in Table (b). To a good accuracy, these values are given by integer numbers as expected.

(a) Chern numbers at $\vartheta = \frac{10}{57}$.

| $Ch_J$ | Gap $L1$ ● | Gap $L2$ ● | Gap $0$ ● | Gap $U1$ ● | Gap $U2$ ● | Gap $U3$ ● |
|---|---|---|---|---|---|---|
| $Ch_\emptyset$ | 0.03077870 | 0.06155740 | 1.00000000 | 1.03077870 | 1.06155740 | 1.09233610 |
| $Ch_{t_1 t_2}$ | -0.99999991 | -1.99963888 | 0.00000000 | 0.99999435 | 1.99998349 | 2.99508127 |
| $Ch_{t_1 u_1}$ | 0.17543859 | 0.35084663 | -0.00000000 | 0.17543819 | 0.35087549 | 0.52593783 |
| $Ch_{t_1 u_2}$ | -0.00000000 | -0.00000003 | -0.00000000 | -0.00000002 | -0.00000004 | -0.00000006 |
| $Ch_{t_2 u_1}$ | -0.00000002 | -0.00000003 | 0.00000000 | -0.00000002 | -0.00000004 | -0.00000005 |
| $Ch_{t_2 u_2}$ | 0.17543858 | 0.35084663 | 0.00000000 | 0.17543818 | 0.35087549 | 0.52593782 |
| $Ch_{u_1 u_2}$ | 0.00000000 | -0.00000000 | 0.99999993 | 0.99999993 | 0.99999993 | 0.99999993 |
| $Ch_{t_1 t_2 u_1 u_2}$ | -0.99999978 | -1.99938378 | -0.00000000 | -0.99999140 | -1.99998058 | -2.99653455 |

(b) Labels of $K$-theory class in $K_0(\mathcal{A}) = \mathbb{Z}^8$.

| $n_J$ | Gap $L1$ ● | Gap $L2$ ● | Gap $0$ ● | Gap $U1$ ● | Gap $U2$ ● | Gap $U3$ ● |
|---|---|---|---|---|---|---|
| $n_\emptyset$ | $0+1.7\times10^{-8}$ | $0+5.2\times10^{-5}$ | $1-1.2\times10^{-10}$ | $1+7.3\times10^{-7}$ | $1+1.3\times10^{-6}$ | $1+2.0\times10^{-4}$ |
| $n_{t_1 t_2}$ | $-1+8.9\times10^{-8}$ | $-2+3.6\times10^{-4}$ | $0+2.6\times10^{-17}$ | $1-5.6\times10^{-6}$ | $2-1.7\times10^{-5}$ | $3-4.9\times10^{-3}$ |
| $n_{t_1 u_1}$ | $0+3.3\times10^{-8}$ | $0+9.4\times10^{-5}$ | $0-5.7\times10^{-10}$ | $0+1.3\times10^{-6}$ | $0+2.1\times10^{-6}$ | $0+2.8\times10^{-4}$ |
| $n_{t_1 u_2}$ | $0-4.9\times10^{-9}$ | $0-3.5\times10^{-8}$ | $0-3.0\times10^{-10}$ | $0-1.8\times10^{-8}$ | $0-3.6\times10^{-8}$ | $0-6.5\times10^{-8}$ |
| $n_{t_2 u_1}$ | $0-1.7\times10^{-8}$ | $0-3.1\times10^{-8}$ | $0+1.1\times10^{-10}$ | $0-1.9\times10^{-8}$ | $0-4.1\times10^{-8}$ | $0-4.9\times10^{-8}$ |
| $n_{t_2 u_2}$ | $0+2.4\times10^{-8}$ | $0+9.4\times10^{-5}$ | $0-5.3\times10^{-11}$ | $0+1.3\times10^{-6}$ | $0+2.1\times10^{-6}$ | $0+2.8\times10^{-4}$ |
| $n_{u_1 u_2}$ | $0+6.0\times10^{-11}$ | $0-5.7\times10^{-10}$ | $1-7.0\times10^{-8}$ | $1-6.9\times10^{-8}$ | $1-7.0\times10^{-8}$ | $1-6.7\times10^{-8}$ |
| $n_{t_1 t_2 u_1 u_2}$ | $-1+2.2\times10^{-7}$ | $-2+6.2\times10^{-4}$ | $0-3.4\times10^{-10}$ | $-1+8.6\times10^{-6}$ | $-2+1.9\times10^{-5}$ | $-3+3.5\times10^{-3}$ |

Let us now inspect the gaps in between the aligned states in the upper block ($U$) and anti-aligned states in the lower block ($L$). There are several gaps at finite $\vartheta$, which remain opened but narrow for $\vartheta \to 0$. We refer to those as being connected to the adiabatic limit. The gaps between the lowest energy bands in each block are the widest, and some of them have been marked with dots and numbered from bottom to top for further examination. Besides these gaps, some gaps open near $\vartheta = 1/2$, which only remain open over a short interval of finite $\vartheta$. Since those gaps are not connected to the adiabatic limit of smooth textures and are as such not treatable with commutative geometry, we coin them *noncommutative spectral gaps*. To compute all Chern numbers corresponding to the gaps marked with dots, we use the periodic approximating algebra formalism introduced in section 4. We first perform an exact diagonalization of the tight-binding Hamiltonian as above on a supercell of size $N_x = N_y = 57$ for $\vartheta = \frac{10}{57}$ (the corresponding texture is depicted in Fig. 4 (a)) and from the eigenstates we construct Fermi projection operators according to Eq. (117) with Fermi energy $E_F$ inside the chosen gaps. They are representations of projections from the periodic approximate algebra of the noncommutative torus, and we can compute their derivations to evaluate the corresponding Chern numbers as detailed in Eq. (118). The partial derivatives $\partial_{\varphi_j} P_F^\varphi$ are numerically implemented as a difference quotient with $\Delta\varphi_j = 10^{-8}$. The results are displayed in Table 3 (b). From these Chern numbers, we can also immediately extract the $\vartheta$-independent coefficients of the Fermi projection with respect to the $K_0$ group generators by using equations (91)-(98). This gives us a full set of integer labels for the gaps as displayed in Table 3 (c).

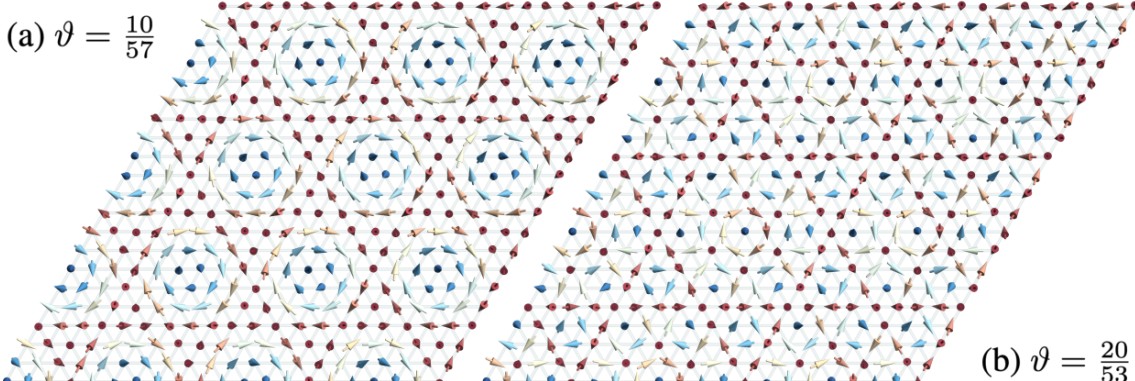

Figure 4: **A skyrmionic multi-$q$ texture at $\vartheta = 10/57$ and $\vartheta = 20/53$**. The figure shows $20 \times 20$ parts of multi-$q$ textures with deg $\hat{\mathbf{n}} = 1$ ($\varphi = 0$, $m = 0$) for $\vartheta = 10/57$ and for $\vartheta = 20/53$ at which unusual band gap formations were noticed in the DOS plots of figure 5. While it is difficult to imagine that the texture in (b) still represents something like a skyrmion lattice, the electronic real-space winding number is indeed quantized.

It can be seen that the numerically computed coefficients are indeed quantized to integer values to good accuracy, and the Chern numbers and coefficients of the zero energy gap (Gap 0 ●) match our preliminary consideration $n_{\emptyset} = n_{\{u_1,u_2\}} = 1$ and 0 for the remaining Chern coefficients. The $\emptyset$ and $\{u_1, u_2\}$ Chern labels of the marked gaps above zero energy ($U$) are equal to 1, while the marked gaps below zero energy ($L$) exhibit 0. This raises the question which electronic band is responsible for carrying this quantized Chern number. Similarly, some bands above the highest of our marked gaps need to carry a Chern label contribution of 1 to $n_{\emptyset}$ and $-1$ to $n_{\{u_1,u_2\}}$ to be consistent with our assessment from general principles. The $\{t_1, t_2\}$ Chern label of the marked gaps above zero energy increases by 1 per gap, which means that the energy bands in between carry a contribution of 1 each. Below zero energy, the sign is flipped. Since we know that the Chern label of the zero energy gap is 0, the contributions of these energy bands need to be compensated by some bands in the same block. The same is true for the block above zero energy. All remaining mixed space first Chern numbers are zero for all marked gaps. Lastly, the $\{t_1, t_2, u_1, u_2\}$ Chern labels of the marked gaps are similar to the $\{t_1, t_2\}$ Chern labels, but have a negative sign below and above zero energy. Also, some compensating energy bands need to exist here.

To investigate further which energy bands exactly contribute to the main Chern numbers, we compute them in dependence on the Fermi energy on a supercell of size $N = 57$ for $\vartheta = 10/57$ as well as $N = 53$ for $\vartheta = 20/53$, as shown in Fig. 5. The density of states in the middle is computed with the KPM on a system of size 400 and the same Hamiltonian parameters as before. This way, we can determine which states carry the quantized $\{u_1, u_2\}$ Chern coefficient contributions. Starting at the bottom of the energy range, the first real-space Chern number $Ch_{\{u_1,u_2\}} = n_{\{u_1,u_2\}}$ is consistently 0, even outside of gaps, until we get close to $E \approx -3$, at which point it jumps to 1 and stays again stable until $E \approx 7$, where it jumps back two 0. This implies, that a single band at these energies carries the entire block's contribution of $\pm$deg $\hat{\mathbf{n}}$. The two energies are correlated with an increased density of states. Indeed, the DOS shows a pronounced peak persisting in the limit of $\vartheta \to 0$, consistent with the *van Hove singularities* (VHS) of the triangular electronic lattice, located at the respective energies $E_{VHS} \approx 2\lambda_{\text{hop}} + \lambda_{\text{xc}} = 7$ and $E_{VHS} \approx 2\lambda_{\text{hop}} - \lambda_{\text{xc}} = -3$ [8].

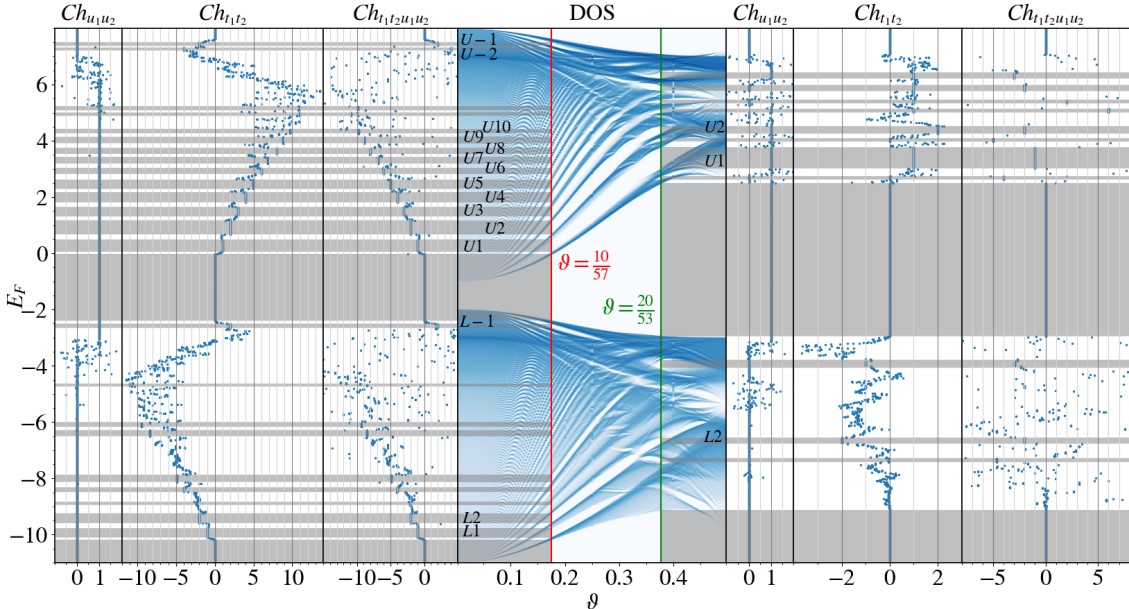

Figure 5: **Main Chern numbers of the skyrmion crystal at zero shift and magnetization.** At $\vartheta = \frac{10}{57}$ and $\vartheta = \frac{20}{53}$ we evaluate the Chern number in systems of size 57 and 53, respectively, for Fermi energies in the full energy range.

Further, one can observe that the first momentum space Chern number $Ch_{\{t_1,t_2\}} = n_{\{t_1,t_2\}}$ of the upper, spin-aligned block of energy states, increases by 1 with each gap connected to the adiabatic limit of $\vartheta \to 0$. For $\vartheta = 10/57$ at the present numerical accuracy, this is well discernible for this block's lowest 10 gaps $U1, \ldots, U10$. This trend seems to reverse at higher energies, and around $E = 7$ the overall Chern numbers become negative. The Chern labels of the highest two gaps $U-2$ and $U-1$ are perceivable to be $-4$ and $-2$, respectively. The gaps are less pronounced for the lower block of spin anti-aligned energy states, but the Chern numbers behave the same with a flipped sign and turning point around $E = -3$. This is consistent with the known findings from mapping the bands to Landau levels which one finds to contribute 1 to the Chern number when below and 2 when above the VSH [8].

For $\vartheta = 20/53$, the gap formation is even less pronounced, and several gaps are not connected to the limit of smooth textures, $\vartheta \to 0$. They seem to behave differently from the rules we observe for the connected gaps. While for $\vartheta = 10/57$, the Chern numbers follow a rather predictable pattern from gap to gap, the same cannot be said for $\vartheta = 20/53$, where no clear pattern can be identified. A look at the real-space distribution of the spins in Fig. 4 (b) reveals a texture that does not resemble a skyrmion lattice at first sight. Yet, the electrons' real-space winding number $Ch_{\{u_1,u_2\}}$ still exhibits quantization. The skyrmionic nature of the texture can thus only reveal itself through (approximate) ergodicity.

In conclusion, we can deduce from the results of this subsection that the bands at the VHS carry the whole $\pm\deg \hat{\mathbf{n}}$ contribution to the first real-space Chern number of the whole block, as well as the whole Chern number contribution to the first momentum space and the second mixed space Chern numbers to compensate for all the other bands' contributions of $\pm 1$ and $\pm 2$. Electrons at the VHS are characterized by a vanishing group velocity. Their spin can thus align perfectly with texture $\hat{\mathbf{n}}$. Notably, those seem to be the only states with nontrivial spin winding.

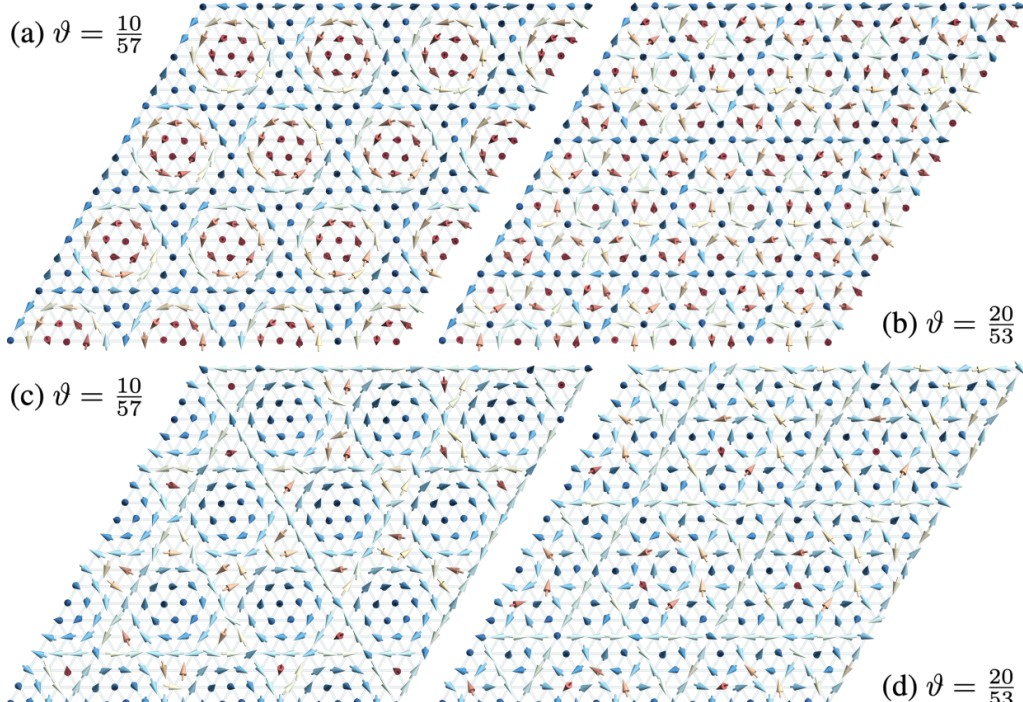

Figure 6: **Skyrmionic multi-$q$ textures for different parameters**. The figure shows $20 \times 20$ parts of multi-$q$ textures with deg $\hat{\mathbf{n}} = -1$ ($\varphi = \pi$, $m = 0$) in (a) and (b) and deg $\hat{\mathbf{n}} = -2$ ($\varphi = 0$, $m = 0.7$) in (c) and (d) for $\vartheta = 10/57$ and for $\vartheta = 20/53$, respectively. The shift in $\varphi$ in (a) and (b) inverts the spins' directions and changes the relative position of the magnetic unit cell in comparison to fig. 4. The magnetisation parameter $m$ raises the spins in $z$-direction. In (c) the winding deg $\hat{\mathbf{n}} = -2$ is perceptible in the sharp corners of the rhombic magnetic unit cell.

## 5.2 Topological phase transitions following a change of real-space winding

In section 2, we demonstrated that the winding number deg $\hat{\mathbf{n}}$ changes discontinuously when varying the texture parameters $\varphi$ and $m$ as displayed in Fig. 2, reproducing the result from [6]. Some example textures for different deg $\hat{\mathbf{n}}$ are presented in Fig. 6. According to our preliminary consideration $Ch_{\{u_1,u_2\}} = n_{\{u_1,u_2\}}$ of the electronic spectrum must change in correspondence. In this section, we want to inspect this case further and investigate how the electronic topology will respond to a change in the real-space topology. We first compute the main Chern numbers of the full Fermi energy range for two different areas of the texture's topological phase diagram and then we compute the change of the Chern numbers over a continuous path in parameter space.

In Fig. 7, we begin with the case $\varphi = \varphi_1 + \varphi_2 + \varphi_3 = \pi$ and $m = 0$, for which we get deg $\hat{\mathbf{n}} = -1$. This change in parameters compared to the previous section is equal to the time-reversal operation (up to translation) as it switches the overall sign of the entire spin texture. All other parameters are the same as for Fig. 5 and, thus, the density of states is also the same as in Fig. 5 for $\varphi = 0$. This is expected as the time-reversal operation does not affect the energy eigenvalues. Accordingly, the second mixed space Chern number remains unaltered as well. This is expected for the same reason, since $Ch_{\{t_1,t_2,u_1,u_2\}} = n_{\{t_1,t_2,u_1,u_2\}}$ is the coefficient of the quadratic term of the integrated density of states in dependence of the length scale parameter, see Eq. (100). On the other hand, both first Chern numbers switch their sign. This is not surprising in the case of $Ch_{\{u_1,u_2\}}$, since it corresponds to deg $\hat{\mathbf{n}}$ and therefore it is odd under the time reversal operation. The same holds true for $Ch_{\{t_1,t_2\}}$.

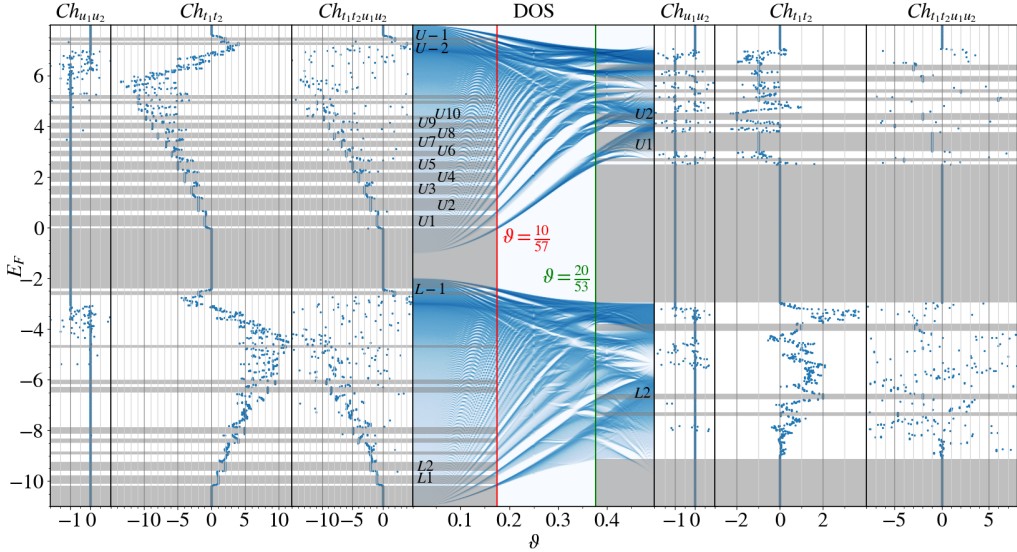

**Figure 7: Main Chern numbers of the skyrmion crystal at $\pi$ shift and zero magnetization.** At $\vartheta = \frac{10}{57}$ and $\vartheta = \frac{20}{53}$, we evaluate the Chern number in systems of size 57 and 53, respectively, for Fermi energies in the full energy range.

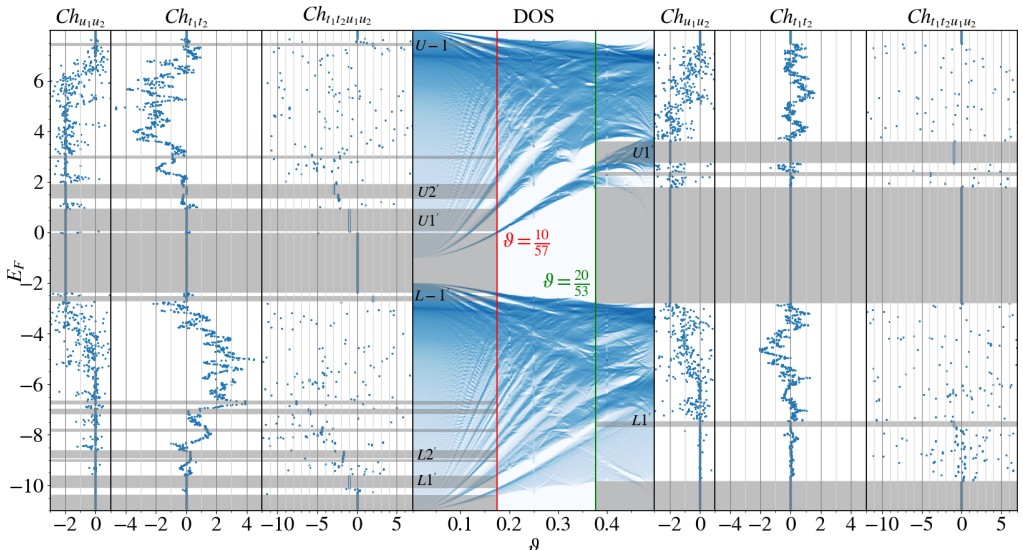

**Figure 8: Main Chern numbers of the skyrmion crystal at zero shift and 0.7 magnetisation.** At $\vartheta = \frac{10}{57}$ and $\vartheta = \frac{20}{53}$ we evaluate the Chern number in systems of size 57 and 53, respectively, for Fermi energies in the full energy range.

Next, we consider the case of $\varphi = \varphi_1 + \varphi_2 + \varphi_3 = 0$ and $m = 0.7$, for which we get deg $\hat{\mathbf{n}} = -2$, displayed in Fig. 8. Here, the texture is not obtained from a global symmetry operation and the result is less predictable from general considerations. At this point in parameter space, only the lowest gaps of each block are sufficiently clear, which gives us less energy range at which our Chern number calculations converge to integers. $Ch_{\{u_1,u_2\}}$ is far less stable than before, but again starts at 0 at the lower end of the energy range and jumps to deg $\hat{\mathbf{n}} = -2$ after $E = -3$ until it goes back to 0 after $E = 7$. Thus, it still behaves analogously to the prior cases. However, unexpectedly $Ch_{\{t_1,t_2\}}$ is 0 in the gaps $U1'$, $U2'$, and $U-1'$ as well as in the gaps $L1'$, $L2'$, and $L-1'$, while $Ch_{\{t_1,t_2,u_1,u_2\}}$ is $-1$ in $U1'$, $-3$ in $U2'$, and 2 in $U-1'$.

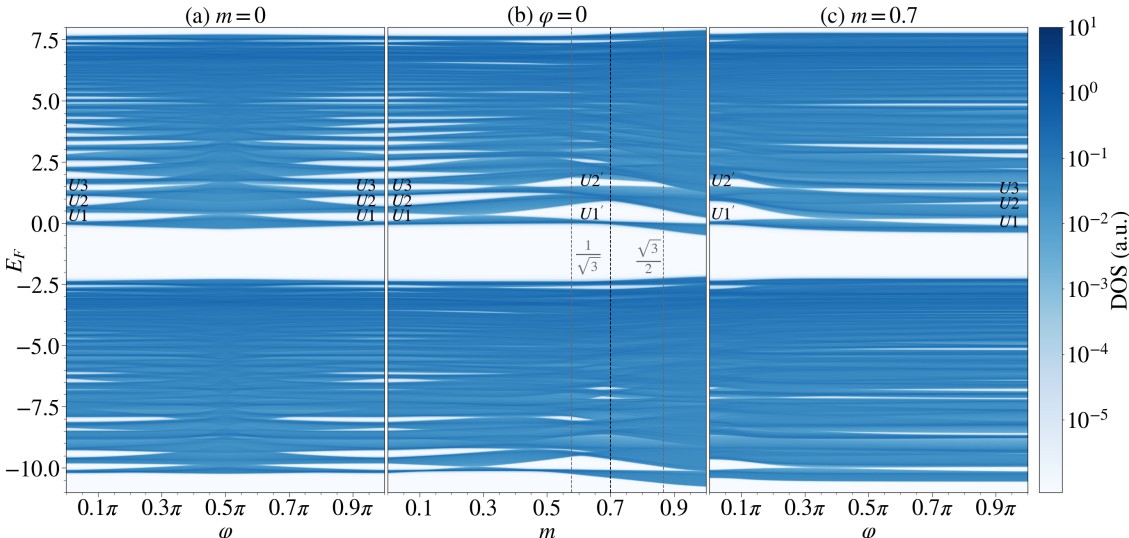

Figure 9: **Density of states evolution across real-space phase transitions.** We perform three different line cuts through the phase space spanned by the texture parameters $(m, \varphi)$.

This implies that the first energy band of the block carries a contribution of 1 to the mixed space Chern number, the second band however carries 2, and the last band carries 2 as well. For the lower block, the $L1'$ and $L-1'$ gaps have the same second mixed space Chern numbers as in the upper block, but in the $L2'$ gap this Chern number does not converge properly.

To understand what happens between these three points in parameter space which are characterized by different winding numbers of the magnetic texture, we focus next on the change of the spectrum and the change of the main Chern numbers in gap $U1$. In Fig. 9 we compute the spectrum of the tight-binding Hamiltonian with the KPM in a supercell of size 400 for $\vartheta = \frac{10}{57}$ and parameters as above for phase shifts $\varphi$ between 0 and $\pi$ at $m = 0$ and $m = 0.7$, as well as for $m$ between 0 and 1 for $\varphi = 0$. The spectrum along the $m = 0$ line in Fig. 9 (a) is symmetric and returns to its initial shape, as the end point is obtained by the time reversal of the starting point. For $\varphi \to \frac{\pi}{2}$ at which point the winding number deg $\hat{\mathbf{n}}$ jumps from 1 to $-1$ all gaps in between the blocks of aligned and anti-aligned states close and reopen again. The $\varphi = 0$ line in Fig. 9 (b) starts at the same parameter point with deg $\hat{\mathbf{n}} = 1$. At $m = \frac{1}{\sqrt{3}}$ the texture's winding number jumps to deg $\hat{\mathbf{n}} = -2$ and at $m = \frac{\sqrt{3}}{2}$ to deg $\hat{\mathbf{n}} = 0$, however this is not apparent from the spectrum. The $U1$ gap closes at an earlier $m$ value and reopens afterwards to $U1'$, and the $U2$ gap closes and stays closed. This makes the $U2'$ from Fig. 8 a factual continuation of gap $U3$ from Fig. 5, which explains that it carries the second Chern number $Ch_{\{t_1, t_2, u_1, u_2\}} = -3$. For $m \to 1$ all gaps except $U1$ have closed. The last line we follow in Fig. 9 (c) starts with the spectrum which is at $m = 0.7$ in Fig. 9 (b), where the $U1'$ and $U2'$ gaps are wide open and clear in the upper block. At the start we have deg $\hat{\mathbf{n}} = -2$ that jumps to deg $\hat{\mathbf{n}} = -1$ around $\varphi = 0.1\pi$. The $U1'$ gap closes briefly around $\varphi = 0.4\pi$ and the second energy band splits to reopen the $U2$ gap we had before at $m = 0$ and $\varphi = 0$. Also multiple higher gaps open. This is consistent with our expectations as the spectrum at the endpoint is in the same area of the winding number phase diagram from Fig. 2 as the spectrum at $m = 0$ and $\varphi = \pi$.

Next, we examine how the main Chern numbers of the first gap $U1$ or, respectively, $U1'$, which is the gap that stays open for most of the parameter range, change along these paths through the parameter space. For the top row of Fig. 10, we computed the exact diagonalization of the tight-binding Hamiltonian on a supercell of size 57 for $\vartheta = \frac{10}{57}$ and parameters as

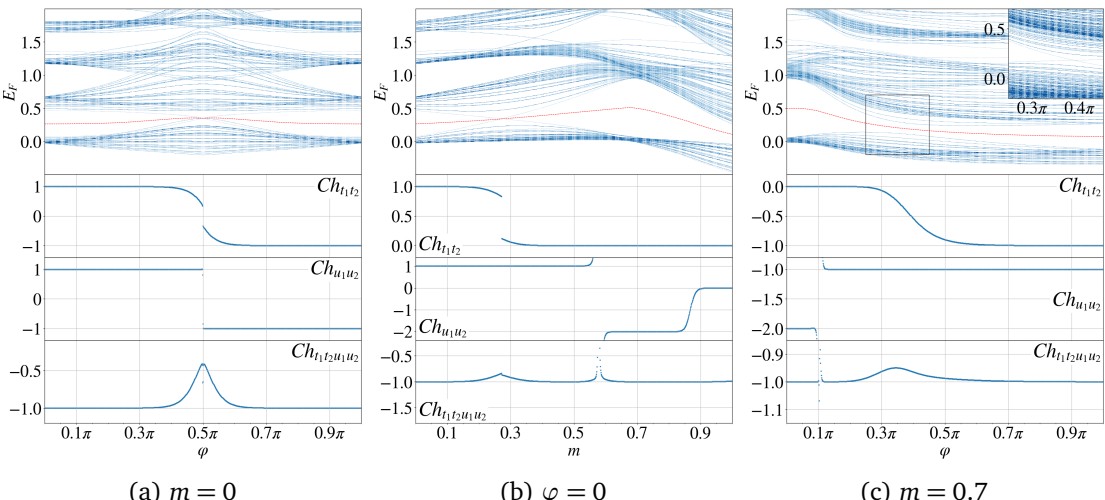

Figure 10: **Chern number evolution across real-space phase transitions.** We perform three line cuts through the phase space spanned by the texture parameters $(m, \varphi)$. We compute the main Chern numbers of the first gap by fixing the IDS to $1 + 1 \cdot \left(\frac{10}{57}\right)^2$ in a system with $57 \times 57$ sites and $\vartheta = \frac{10}{57}$, the red dashed line marks the Fermi energy positioned in the middle of the gap. The inset in subfigure (c) displays the energy eigenvalues computed in a larger system with $137 \times 137$ sites and $\vartheta = \frac{24}{137}$.

above and plotted the individual energy eigenvalues. From this diagonalization, we also computed the corresponding main Chern numbers. Since we are particularly interested in their response to the discontinuities in the texture, we choose a phase space origin position $\varphi$ to move the singularity onto the site at $\boldsymbol{x}_0$. For the path along $m = 0$ and $m = 0.7$ we choose $[\pi + \arccos\frac{m}{\sqrt{3}} + \varphi, \pi + \arccos\frac{m}{\sqrt{3}}, -2\arccos\frac{m}{\sqrt{3}}]$ and for $\varphi = 0$ we choose $[\pi, \pi, 0]$.

Along the $m = 0$ line, depicted in Fig. 10 (a), we have a sharp jump from 1 to $-1$ in $Ch_{\{u_1, u_2\}}$ at $\varphi = \frac{\pi}{2}$, where the winding number switches from 1 to $-1$ as well. $Ch_{\{t_1, t_2\}}$ follows through, but the transition is less sharp and nearly continuous over a wider parameter range. It coincides with two states crossing the first gap. The second Chern number $Ch_{\{t_1, t_2, u_1, u_2\}} = -1$ remains mostly constant but undergoes a peak in the vicinity of the winding number transition. Similarly, following the $\varphi = 0$ line in parameter space, shown in Fig. 10 (b), $Ch_{\{u_1, u_2\}}$ matches the winding number of the texture. The first jump from 1 to $-2$ at $m = \frac{1}{\sqrt{3}}$ coincides with a singularity of $Ch_{\{u_1, u_2\}}$, while the second transition from $-2$ to 0 at $m = \frac{\sqrt{3}}{2}$ is continuous. This is caused by the real-space position of the lattice sites and whether the singularity of the magnetization vector field occurs close by one of them. If we would additionally average over the phase space, the Chern number would exhibit a singularity at both parameter points. However, $Ch_{\{t_1, t_2\}}$ does not follow suit with the winding number and changes at a lower value of $m$: already at $\approx 0.27$ a band crossing in the first gap occurs and $Ch_{\{t_1, t_2\}}$ changes nearly continuously from 1 to 0. The second Chern number $Ch_{\{t_1, t_2, u_1, u_2\}} = -1$ remains again mostly constant but undergoes a peak in the vicinity of the band crossing and a singularity at the point of the singularity of $Ch_{\{u_1, u_2\}}$ however, does not respond to the second continuous change of $Ch_{\{u_1, u_2\}}$. The behavior of the Chern numbers along the $m = 0.7$ line in Fig. 10 (c) holds more unexpected features. While $Ch_{\{u_1, u_2\}}$ exhibits a singularity around $\varphi = 0.1\pi$ and jumps from $-2$ to $-1$ following the winding number, $Ch_{\{t_1, t_2\}}$ changes continuously over a wide range around $\varphi = 0.4\pi$ from 0 to $-1$, while none of the eigenvalues in the system of size 57 crosses the gap. However, for a change in Chern number to occur in a system there must be a gap closing [69] in the spectrum, and indeed for a system size of 137, we observe a crossing in the

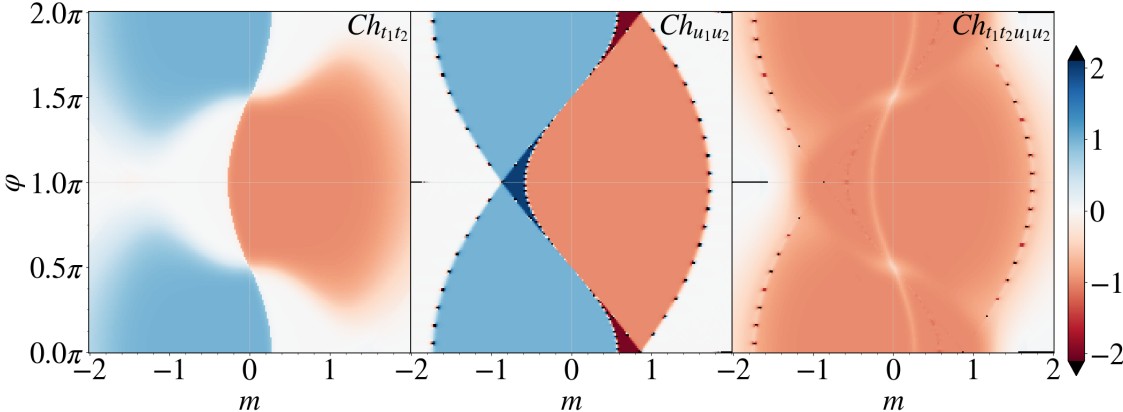

Figure 11: **Topological phase diagrams.** For constant IDS $= 1 + 1 \cdot \left(\frac{10}{57}\right)^2$, these figures trace the Chern numbers across the phase space spanned by $(m, \varphi)$. Notably, $\mathrm{Ch}_{u_1 u_2}$ is in one-to-one correspondence with the phase diagram obtained through $\deg \hat{\mathbf{n}}$ of the texture. $\mathrm{Ch}_{t_1 t_2}$ has a pronounced correlation with the net-magnetization of the real-space texture, while $\mathrm{Ch}_{t_1 t_2 u_1 u_2}$ seems to be responsive to both of these qualities of the real-space texture.

eigenvalues. At this point we need to remark, that this is not purely dependent on the system size, but rather on the ergodicity condition. For some sizes the crossing does appear for others it does not. This could also be resolved by computing the phase space average. Again for this path, the second Chern number $Ch_{\{t_1, t_2, u_1, u_2\}} = -1$ remains mostly constant with a peak around the band crossing and singularity at the singularity position of $Ch_{\{u_1, u_2\}}$.

Since the parameter position of the band crossing at which $Ch_{\{t_1, t_2\}}$ changes does not coincide with the parameter position of the jumps in winding number, except for at $m = 0$, we further scan over the whole relevant area of parameter space to identify its pattern. As displayed in Fig. 11, we compute the main Chern numbers for the tight-binding Hamiltonian on a system of size 57 with parameters as above for $\varphi$ ranging from 0 to $\pi$ and $m$ ranging from $-2$ to 2 and a phase space origin set to $\varphi = [\varphi, 0, 0]$.

Here, $Ch_{\{u_1, u_2\}}$ perfectly reproduces the winding number phase diagram from Fig. 2, which we already observed for the three paths. On the border lines between the different areas, it exhibits singularities at regular distances. Those coincide with the points in parameter space for which a discontinuity in the magnetization texture lies in the immediate vicinity of a lattice site. Due to the different choices of $\varphi$ these positions do not coincide with the singularities observed along the paths above. Computing the average over the phase space would cause singularities along the whole border. Interestingly, the sharp change in $Ch_{\{t_1, t_2\}}$ at which band crossings occur coincides with the parameter line in which the net magnetization $\langle \hat{n}_z \rangle$ is zero. Precisely at those positions, the system is time-reversal invariant (modulo lattice symmetry), and thus $Ch_{\{t_1, t_2\}}$ is forced to vanish by reasons of symmetry. The borders of the areas with $Ch_{\{t_1, t_2\}} = \pm 1$ for $|m| \to 2$ resemble the areas of $Ch_{\{u_1, u_2\}}$ but are more fuzzy which might be a finite system size effect. As for the paths before, the second Chern number $Ch_{\{t_1, t_2, u_1, u_2\}} = -1$ remains mostly constant. At the transition borders of the other Chern numbers it is affected by peaks or singularities but does not change, except for the outside border for $|m| \to 2$ for which it transitions to 0.

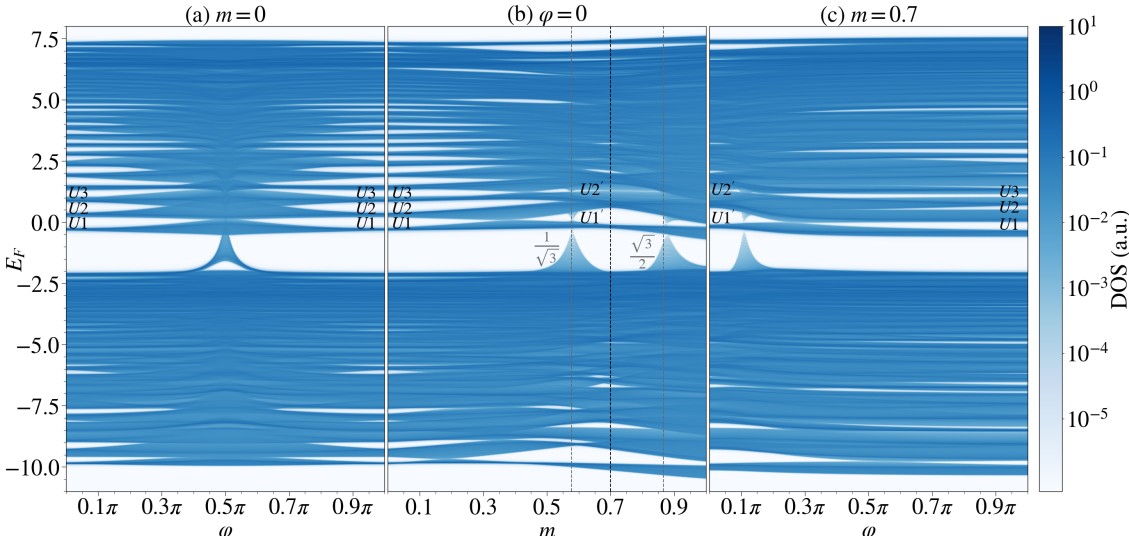

Figure 12: **Density of states evolution across real-space phase transitions with regularization.** We perform three different line cuts through the phase space spanned by the texture parameters $(m, \varphi)$. The texture is regularized with $\eta = 0.1$.

## 5.3 On the existence of topological spectral flow

While all transitions of $Ch_{\{t_1, t_2\}}$ are accompanied by a band crossing, we do not observe band crossings that correspond to the change in $Ch_{\{u_1, u_2\}}$, especially no closing of the zero energy gap. This is due to the singular nature of the transition as discussed in section 2.2. To make the transition continuous, we regularize the singularity with a parameter $\eta = 0.1$ as described in Eq. (8). This leads to a shift in energy close to the transition, a shift of the critical parameter point $(\varphi_c, m_c)$ at which the transition occurs, and a crossing of energy eigenstates. We compute the spectrum of the tight-binding Hamiltonian with regularized magnetic texture analogous to Fig. 9 with the KPM on a system of size 400 as displayed in Fig. 12.

Indeed, along the $m = 0$ path in Fig. 12 (a) we clearly see the band crossing at $\varphi = \pi$. The bands of the upper block bend downwards and the bands of the lower block bend upwards. At the top of the lower block, a single band bends upwards until it touches the bottom band of the upper block, crossing the zero energy gap. Similarly, in Fig. 12 (b) we also see band crossings along $\varphi = 0$ at the parameter positions of the winding number jumps, however with a lower density of states. Since the phase space origin $\varphi$ of the texture for this computation was chosen in a way, that the first former singularity would be sitting right on the lattice site with label **0**, the first band crossing fully closes the gap while the second crossing leaves a small gap open. In Fig. 12 (c), we choose $\varphi$ to position the former singularity on the origin lattice site as well. Also here, we can see how states from the lower block bend upwards and cross the gap around $0.1\pi$ where the winding number transition takes place.

Finally, we need to examine the effect of the regularization on the Chern number transitions, which we do in Fig. 13 computed analogously to Fig. 10, but with the regularization procedure applied. We can observe the behavior of the individual energy eigenstates from the tight-binding Hamiltonian for system size 57 and the corresponding main Chern numbers. Along $m = 0$ in Fig. 13 (a), we see how a high number of states is bent upwards such that several states close the gap. The transition of the Chern numbers is nearly identical to that in Fig. 10 (a), except for $Ch_{\{t_1, t_2, u_1, u_2\}}$ which exhibits a broader peak than before. In Fig. 13 (b), we see a single energy state close the gaps for the first transition, due to the choice of the phase space origin, and a few states bend only halfway upwards to close the gaps for the sec-

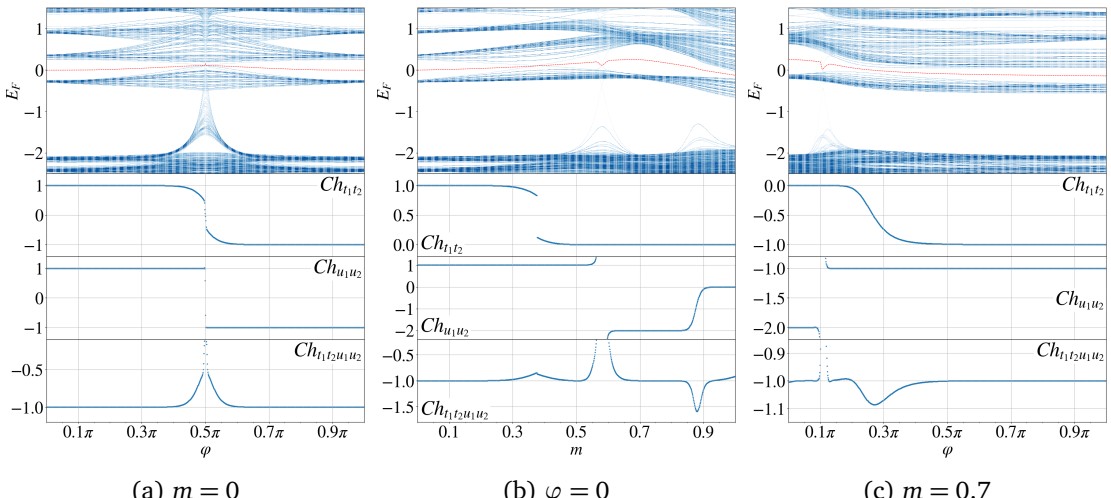

|     |     |     |
| --- | --- | --- |
| (a) $m = 0$ | (b) $\varphi = 0$ | (c) $m = 0.7$ |

Figure 13: **Chern number evolution across real-space phase transitions with regularization.** We perform three different line cuts through the phase space spanned by the texture parameters $(m, \varphi)$. We compute the main Chern numbers of the first gap by fixing the IDS to $1 + 1 \cdot \left(\frac{10}{57}\right)^2$ in a system with $57 \times 57$ sites and $\vartheta = \frac{10}{57}$, the red dashed line marks the Fermi energy positioned in the middle of the gap.

ond transition. The Chern numbers behave very similarly to those in Fig. 13 (b), except for the position of the $Ch_{\{t_1, t_2\}}$ transition which happens at around $m \approx 0.375$ in comparison to $m \approx 0.275$ for the texture which was not regularized. The regularization predominantly impacts the magnetization and therefore also the parameter position for which $\langle \hat{n}_z \rangle$ goes to zero. $Ch_{\{t_1, t_2, u_1, u_2\}}$ exhibits a new dip at the second transition. In Fig. 13 (c), we also only have a single energy state which nearly closes the gap at the position where formerly the singularity occurred. The position of the $Ch_{\{t_1, t_2\}}$ transition is shifted from around $\varphi = 0.4\pi$ to around $\varphi = 0.27\pi$, which is also due to the change in magnetization.

The reason why the spectral flow [70] over the phase transition does or does not appear in real-space can be found in algebraic arguments. Namely, the observable algebra $M_2(\mathbb{C}) \otimes (C(T^2) \rtimes_\theta \mathbb{Z}^2)$ itself requires the continuity of the magnetic texture as a function of the phase space $T^2$, which also implies continuity on the real-space $\mathbb{R}^2$. However, it is impossible to keep the texture normalized to a field of unit vectors $\hat{\mathbf{n}}(\mathbf{x}) = \mathbf{n}(\mathbf{x})/\|\mathbf{n}(\mathbf{x})\|$ and maintain continuity across the transition. There must exist a critical point along the reaction path at which the texture exhibits a discontinuous singularity somewhere in real-space. With this point, we no longer have a continuous path of Hamilton operators, which at each point are given by the covariant representation of an element from the observable algebra. Therefore, the Fermi projections before and after the critical point are no longer homotopic within the space of projections from the observable algebra, even without closing the spectral gap, and the gap labels no longer need to be preserved.

This point of zero measure will stay unnoticed by a generic computation, as it will almost surely fall somewhere in between the lattice sites and thus go unnoticed. And if it were to fall on a lattice site, the program would crash due to the undefined behavior. Once the texture singularity is regularized, the texture can remain continuous along the reaction path, and the $K$-theory of the observable algebra enforces the existence of the spectral flow: the continuous path of the Fermi projections must be interrupted by a gap closure to allow the gap labels to be changed.

# 6   Discussion

In this work, we provide a $C^*$-algebraic view on the interaction of real- and reciprocal space topology in skyrmion crystals. As first shown in [10], and elaborated here, the electronic observables of multi-$q$ textures form a $C^*$-algebra $\mathcal{A}_\theta$, known as the noncommutative torus. For the case of a skyrmionic 3$q$-state hosted by a two-dimensional triangular lattice [8, 9], we have determined the precise equivalence class of projections in $K_0(\mathcal{A}_\theta) = \mathbb{Z}^8$ for the main series of spectral gaps. To a good accuracy, a numerical computation of noncommutative Chern numbers for a given spectral gap indeed produces a series of eight integers, identifying the respective class in $K_0(\mathcal{A}_\theta)$: the first main result of this work.

By observing the dependence of Chern numbers as a function of the Fermi energy, one observation stands out: the Chern number describing the real-space winding of the electronic spin is carried by electrons at the van Hove singularity. They are characterized by a vanishing group velocity likely facilitating the perfect alignment of the electronic spin with the imposed magnetic texture. Most bands producing a quantized anomalous Hall response have a trivial real-space winding number. This observation might challenge some possible preconceptions about the physical origin of the Hall response.

Inspired by the work [6], we set out to study the evolution of electronic states across various topological phase transitions of the magnetic texture. A first idea of the physics of noncollinear textures is typically obtained through the language of emergent electromagnetism. Our results indicate that this picture can only serve as a first guess into the adiabatic behavior, but cannot account for the true complexity of the electronic spectrum away from the limit of strong coupling and smooth textures. We could not identify a simple rule that ties the quantized Hall response to the electrons' real-space winding. A change in a band's quantized anomalous Hall response also occurs away from the real-space transitions and generally coincides with a spectral flow between the neighboring, participating bands. Whether or not spectral flow occurs in the electronic system as one passes through the real-space transitions depends not only on the topological labels but also on the nature of the critical point itself. Since it is impossible to maintain continuity of the texture in passing from one topological configuration to another without sacrificing spin normalization, this leaves the door open for a brief exit out of the observable algebra at which the change of the electronic topology can occur unbeknownst to the observer. A real-world possibility is that the texture is not truly ergodic. In this case, a Bloch point singularity in the magnetic textures can occur in between lattice sites, and no spectral flow would be visible in the experiment. Ultimately, this question could be treated experimentally in the future: whether or not spectral flow is visible will hinge on the details of the magnetic configurations close to the phase transition.

Since the formalism naturally incorporates the possibility of applied magnetic fields and multi-$q$ textures are often stabilized with the help of such external fields, we see this as a natural next step for our investigation into the relation between noncommutative geometry and noncollinear magnetism. Recently, antiferromagnetic multi-$q$ textures have been proposed as another route toward realizing topological electronic states via complex magnetic order. Fundamentally, no change in the observable algebra is needed. It is still described by the same noncommutative torus and we therefore see this as another fruitful application for our formalism. Lastly, the dynamical aspect should not be overlooked as spin and charge currents adiabatically induced by time-dependent magnetism have close connections to the underlying electronic topology [65]. Overall, we believe that our work showcases the usefulness of $C^*$-dynamical systems such as the noncommutative torus in the description of magnetic multi-$q$ order, where they can assist in disentangling the complex electronic structure and the associated topological features.

# Acknowledgments

We thank Juba Bouaziz, Gregor Michalicek, and Tom Stoiber for fruitful discussions.

**Funding information** Funding was received under the Deutsche Forschungsgemeinschaft (DFG, German Research Foundation) – TRR 173/3 – 268565370 (project A11), and TRR 288/2 – 422213477 (project B06). We acknowledge funding under SPP 2137 "Skyrmionics". This work was further supported by the Max Planck Graduate Center with the Johannes Gutenberg-Universität Mainz (MPGC). We also gratefully acknowledge the Jülich Supercomputing Centre and RWTH Aachen University for providing computational resources under projects jiff40 and jara0062. E.P acknowledges support from the U.S. National Science Foundation through the grants DMR-1823800 and CMMI-2131760 and from the U.S. Army Research Office through contract W911NF-23-1-0127.

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
