# Peer review of "A $C^\ast$-algebraic view on the interaction of real- and reciprocal space topology in skyrmion crystals"

_SciPost Physics Core, doi:SciPost Phys. Core 7, 080 (2024)_

## Round 2 · Referee Report · Anonymous (Referee 1) · 2024-7-8

Strengths

See report

Weaknesses

See report

Report

\documentclass{article}
\begin{document}

This manuscript is a longer version of Ref.\ [10]:
F. R. Lux, S. Ghosh, P. Prass, E. Prodan and Y. Mokrousov,
\textit{Unified topological characterization of electronic states
in spin textures from noncommutative K-theory},
Phys. Rev. Res. 6, 013102 (2024),
doi:10.1103/PhysRevResearch.6.013102.

The differences with Ref.\ [10] are
(i) a more mathematical treatment of the formalism presented in Ref.\ [10]
(ii) and new results that are relevant to
noninteracting electrons hopping on
a triangular lattice in the background of a magnetic texture realizing
a 3\bmq hexagonal skyrmion lattice.

The questions addressed in this paper are the following.
Consider noninteracting electrons hopping on a Bravais lattice
with hopping amplitudes preserving the crystalline symmetries of the lattice
such that they couple to an effective site-dependent magnetic vector field
\bmn through an effective Zeeman term with a periodicity distinct
from that of the lattice.
Two questions are solved in this manuscript.
First,
is the background magnetic field
\bmn topologically nontrivial and can the eigenstates of the
noninteracting Hamiltonians be organized into bands''
with topological attributes?
Second,
what are the consequences of a parametric change of
\bmn that changes the topology of \bmn
for the topological spectral properties of the electrons?
Both questions are answered numerically by computing the electronic spectra
using exact diagonalization techniques in a given background of
the background magnetic field \bmn in Section 5. These answers are
new to the best of my knowledge.

The main results of Secs.\ 2-3 are Eqs.\ (83)-(90) or, equivalently,
Eqs. (91)-(98), in which eight distinct Chern numbers are computed
for electrons hopping with nearest-neighbor hopping on the sites of
a triangular lattice in the background of an effective magnetic field
realizing a skyrmion hexagonal lattice.
Section 4 explains how the eight Chern numbers in Eqs.\ (83)-(90)
are to be approximated for finite lattices
obeying periodic boundary conditions.
Section 5 presents the novel computations of these eight Chern numbers
and discusses their dependencies upon changing the background
effective magnetic field.

I recommend publication of this paper in SciPost. I leave it to the authors
to address the following comments in a revised version of their manuscript
if they choose so.

Comments:

01) I felt that Teo and Kane
https://arxiv.org/abs/1006.0690
deserve some credit for the idea
that spectral properties of noninteracting fermions are sensitive to both the
topology of the torus associated to twisting boundary conditions and
the topology of defective background fields.

02) Do I understand correctly that r in Eq.\ (5)
is defined by the range of \bmm\nathhbZd
such that the Fourier coefficient \bmn\bmG\bmm
is either zero or smaller than some threshold. As such, it
can be larger than the dimensionality d of space.
Perhaps this could be emphasized around Eq.\ (5).

03) I believe that \bml belongs to Zd not Z
after Eq. (16).

04) Does not the introduction of the r-torus presume that
\bmqi with i=1,,r are all linearly independent,
which, however, is not necessarily the case?
Would it not be cleaner to introduce
˜r before Eq. (18) instead of after Eq.\ (18)?

05) Should it not be \bmx as the argument
of ^\bmn instead of \bmω(\bmx) in the first
equality of Eq.\ (26)? Perhaps it could be said that the second quality
of Eq. (26) follows from Eq.\ (17).

06) I would have traded (33) for \bmmα\bmm
whereby old Eq.\ (33) becomes Eq. (34).

07) Should one not set [ti,tj]=0 for i,j=1,,d in Eq.\ (34)
in order to get the group Zd?

08) Before Eq.\ (37), reference is first made to a Banach Algebra''
and then to a complex Banach Algebra''. I am guessing that the adjective
complex refers to the underlying field. Perhaps, over a field''
can be added to the first instance of a Banach Algebra''.

09) Should one not set [uk,ul]=0 for l,l=1,,r in Eq.\ (47)
in order to endow C(Tr) with a Abelian group structure under point-wise
multiplication?

10) Why is p chosen to be a prime number in Eq. (102)? Is it because
the normal subgroups should be fields as well?

11) I did not understand the sentence
There are several clear gaps, which are connected to the adiabatic limit ...''

12) Figure 3 contains a Figure in panel (a) and two tables with panels
(b) and (c). It was a little confusing to call Tables in the text.
Perhaps Panel (a) could be made a stand alone Figure, while panels (b)
and (c) could be made stand alone Tables.

\end{document}

Requested changes

See Report

Recommendation

Publish (meets expectations and criteria for this Journal)

  • validity: ok
  • significance: ok
  • originality: ok
  • clarity: ok
  • formatting: good
  • grammar: good

Author:  Pascal Prass  on 2024-10-14  [id 4867]

(in reply to Report 1 on 2024-07-08)

Dear Referee,

We want to thank you for your objective and kind assessment of our work and for the helpful comments, which speak of the time and effort dedicated to the review of our manuscript. Below, we address how the manuscript has been modified as a response to the issues raised.

  1. I felt that Teo and Kane https://arxiv.org/abs/1006.0690 deserve some credit for the idea that spectral properties of noninteracting fermions are sensitive to both the topology of the torus associated to twisting boundary conditions and the topology of defective background fields.

This reference describes a framework for the classification of topological defects of varying dimensionality and their impact on the emergence of protected gapless modes through the lens of the bulk-boundary correspondence. This is closely related to sections 5.2 and 5.3 of our manuscript, in which we observe band crossings while passing a point defect in a three-dimensional space, comprising two spatial dimensions and one parameter dimension, as outlined in section 2.3. We feel it is only appropriate to include this reference and acknowledge the authors' work on this kind of correspondence.

  1. Do I understand correctly that r in Eq. (5) is defined by the range of mZd such that the Fourier coefficient nGm is either zero or smaller than some threshold. As such, it can be larger than the dimensionality d of space. Perhaps this could be emphasized around Eq. (5).

It is correct, that r may exceed the dimensionality d. However, the truncated texture in Eq. (5) is supposed to consist of the lowest order Fourier modes without a restriction on the norm of nGm. In the regime of multi-q textures it is expected that these modes dominate the higher order terms. In the current form of Eq. (5) m may take values in 1,0,1d and thus it is r3d. To be consistent with the r occurring in the trigonometric formulation in Eq. (6), we modified Eq. (5) to

n(x)ri=rnqiei(qix+φi)

with qi=qi, nqi=nqi, and zero mode with q0=0, φ0=0, and nq0R. In this form, r is may take values up to (3d1)/2.

  1. I believe that l belongs to Zd not Z after Eq. (16).

Indeed, it was supposed to read Zd.

4.Does not the introduction of the r-torus presume that qi with i=1,,r are all linearly independent, which, however, is not necessarily the case? Would it not be cleaner to introduce ˜r before Eq. (18) instead of after Eq. (18)?

In Eq. (18) the r-torus is introduced as the ambient manifold of the orbit. Therefore, the linear independence of the qi does not need to be presumed. We agree, however, that it is a suggestive notation that should be amended. We opted to add a paragraph at the end of section 2.1. to introduce ˜r and to address the differences between periodic and aperiodic textures. This way we only need to discern two cases for Eq. (18).

5.Should it not be x as the argument of ˆn instead of ω(x) in the first equality of Eq. (26)? Perhaps it could be said that the second quality of Eq. (26) follows from Eq. (17).

With a misuse of notation, we use the same symbol ˆn for the texture defined on real space and on the phase torus: ˆn=ˆnω. We added an explanation for this identification, which is the continuum version of Eq. (17), to the beginning of section 2.2., and refer to it for Eq. (26).

6.I would have traded (33) for mαm whereby old Eq. (33) becomes Eq. (34).

We added mαm to Eq. (32).

7.Should one not set [ti,tj]=0 for i,j=1,,d in Eq. (34) in order to get the group Zd?

The relator was meant to indicate that [ti,tj] is equal to the group identity in this presentation. For improved clarity, we changed it to the relation titj=tjti.

8.Before Eq. (37), reference is first made to a Banach Algebra and then to a complex Banach Algebra. I am guessing that the adjective complex refers to the underlying field. Perhaps, "over a field" can be added to the first instance of a Banach Algebra.

This is correct. We added "algebra over a field" at the first instance.

9.Should one not set [uk,ul]=0 for k,l=1,,r in Eq. (47) in order to endow C(Tr) with a Abelian group structure under point-wise multiplication?

Same as for comment 7), we changed it to uiuj=ujui.

10.Why is p chosen to be a prime number in Eq. (102)? Is it because the normal subgroups should be fields as well?

p only needs to be a natural number greater than one for this construction to work. The choice of p being a prime number is part of the example and carries no significance. We changed it to the more general assumption.

11.I did not understand the sentence "There are several clear gaps, which are connected to the adiabatic limit ...''

With the adiabatic limit, we refer to systems with ϑ0 for which the slow spatial variation allows for the adiabatic assumption that the electronic states remain in the same energy state. In general, the adiabatic criterium also depends on the coupling parameter λxc, the lifetime of the itinerant electrons, and the Fermi velocity. However, in the strong coupling limit, we can assume that adiabaticity is realized for large skyrmion size.

In the density of states figure Fig. 3(a) there are several gaps, some of which are only open over a short interval of finite ϑ. With gaps connected to the adiabatic limit, we refer to the once, which shrink in width for ϑ0, but only seem to close at ϑ=0. Between some bands of higher density the would-be gap is filled by states with low density. Therefore, with "clear" gaps we meant emphasize that the gaps are free of any states.

For the resubmission of the manuscript rewrote the aforementioned sentence to make clearer to which features of the figure we refer.

12.Figure 3 contains a Figure in panel (a) and two tables with panels (b) and (c). It was a little confusing to call Tables in the text. Perhaps Panel (a) could be made a stand alone Figure, while panels (b) and (c) could be made stand alone Tables.

We agree, separating the figure from the tables improves the layout.

Sincerely,

Pascal Prass and co-authors

---

## Round 3 · Author Response

We are grateful for the opportunity to respond to the referee's comments and to resubmit our manuscript with the appropriate revisions.
Please find attached a list of changes made to the manuscript. Our point-by-point response to the questions and comments raised by the referee has can be found in the report section of the submission page.

Sincerely, and on behalf of all co-authors,

Pascal Prass

---

## Round 3 · List of Changes

References: 1. DOI added to Ref. [63] (former [61]) 2. Ref. [11] and [31] added

Figures: 1. Panels in former Fig. 3 arranged into Fig. 3 and Tbl. 1

Main text: 1. Added context to Eq. 5 2. Additional paragraph at the end of Sec. 2.1 3. Rewritten first paragraph of Sec. 2.2 4. Shortened last paragraph of Sec. 2.4 5. Added context to Fig. 1 6. Changed form of Eq. 74 and 75 7. Added context to Eq. 102 8. Fixed order of operators in Eq. 108 9. Added context to Fig. 3 10. Added context to Tab. 1 11. Added Ref. [11] and [31] with context 12. Small corrections of spelling and typing mistakes

---

## Editorial Decision

published